# Event-related potentials of stimuli inhibition and access in cross-modal distractor-induced blindness

**Sophie Hanke** [ORCID] *, **Michael Niedeggen**

Division General Psychology and Neuropsychology, Department of Education and Psychology, Freie Universität Berlin, Berlin, Germany

* sophie.hanke@fu-berlin.de

**Data Availability Statement:** All data is available now in an open repository by Mendeley Data (https://doi.org/10.17632/93nxgxntbk.1).

**Funding:** The author(s) received no specific funding for this work.

## Abstract

Distractor-induced blindness (DIB) describes a reduced access to a cued visual target–if multiple target-like distractors have been presented beforehand. Previous ERP data suggest a cumulative frontal inhibition triggered by distractors, which affects the updating process of the upcoming target. In the present study, we examine whether the modality of the cue—formerly defined in the visual domain–affects the expression of these neural signatures. 27 subjects were tested in a cross-modal DIB task: Distractors and targets were defined by a transient change of stimuli shape in a random-dot kinematogram. The onset of the target was announced by a rise in amplitude of a sinusoidal tone. Behavioral results confirmed that detection of the target relies on the number of preceding distractor episodes. Replicating previous unimodal results, ERP responses to distractors were characterized by a frontal negativity starting at 100 ms, which increases with an increasing number of distractor episodes. However, the processing–and detection–of the target was not characterized by a more-expressed P3 response, but by an occipital negativity. The current data confirm that the neural signatures of target awareness depend on the experimental setup used: In case of the DIB, the cross-modal setting might lead to a reduction of attentional resources in the visual domain.

## 1. Introduction

On a daily basis, our environment inundates us with an immense quantity of information. However, the human brain cannot process all of this data at once [1]. In order to maintain the effectiveness and functionality of our behavior, a selection process becomes imperative [2]. It has been hypothesized that this process relies on the activation of filters, which have also been described as *attentional sets*, to control what information captures our attention and what information we disregard [3, 4]. Positive attentional sets facilitate our focus on stimuli with specific attributes, while negative attentional sets suppress the processing of stimuli with certain attributes that ought to be ignored [5].

Although these mental filters typically prove highly beneficial, a negative attentional set can lead to the involuntary suppression and/or insufficient processing of target stimuli [6–8]. One

**Competing interests:** The authors have declared that no competing interests exist.

illustration of this can be found in the *negative priming* effect, where previous exposure to a stimulus, intended to be ignored, results in a delayed response when the same stimulus reappears shortly afterwards as a target requiring action [9–11]. Another example provides *distractor-induced blindness* (DIB) [12]: Here, the cortical suppression mechanism affects the probability to detect an upcoming target stimulus [13–16].

The experimental DIB setup–in an adapted form also utilized in the current study—is based on a temporal selection task involving rapid serial visual presentation (RSVP) [17]. When task-irrelevant, target-like distractors are presented in an RSVP-stream, a decreased likelihood of detecting an upcoming target–indicated by a preceding cue–can be observed. Importantly, the number of distractors has been identified as a decisive factor: The higher the number of distractors, the lower the probability to detect a target [13].

A DIB effect can be induced using dynamic visual features as target/distractor episodes, such as transient motion [12, 13, 18, 19] and orientation change [15, 16, 20], as well as visual object features, such as color change [21, 22]. Furthermore, local luminance change has been successfully implemented as a target feature eliciting the blindness effect in a multi-sensory context [22]. Overall, the adaptability of the effect seems to extend to different feature dimensions.

Recent research has revealed the possibility to extend the effect into the domain of auditory perception [22–24]. A transfer of the distractor effect was observed–revealing that the number of auditory distractors also affects the possibility to detect an auditory target. Correspondingly, the effect has been labeled unimodal *distractor-induced deafness* (DID) [23]. Studies where cue and target are defined in the same sensory modality are in the following referred to as 'unimodal'.

Finally, the distractor-induced process can furthermore be found in cross-modal setups, with the term 'cross-modal' referring to the incongruence between the sensory modalities of the cue and target stimulus: In the case of *cross-modal DIB*, multiple visual distractors impair the detection of a visual target, signaled by an auditory cue [22]. In a cross-*modal DID* setup, on the other hand, modalities are reversed. Here, multiple auditory distractors diminish the detection of an auditory target indicated by a visual cue [24]. The limitations section of this manuscript provides an in-depth discussion about the implications associated with this definition of 'cross-modality.'

In summary, the behavioral data across all the above-described modalities are in line with the idea that distractors–independently from feature and modality—cumulatively activate a negative attentional set. Its deactivation–required with the onset of the cue–appears to be a time-consuming or sluggish process [25], and targets presented in temporal proximity to the cue are therefore not available [12]. Whether the cue is defined globally, i.e. in the periphery [26], or locally, i.e. within a central stream [12, 18, 19], does not seem to alter this consequence. Neither does it make a difference if the cue episode is defined as a letter [14], luminance change [21, 27], or the onset of 'transparent' motion [26], instead of the typically implemented color change [12, 17–20, 28]. In previous DIB studies, the cue was thus viewed as a simple temporal marker, that merely has to be highly salient and should not require any elaborate processing, to guide attentional facilitation and inhibition [14]. However, as explained in the following, current research seems to cast this assumption in a new light.

Support for the inhibition account has been observed in a series of ERP studies–mostly based on the visual DIB [17, 20, 21, 26]: Distractor processing was characterized by a frontal negativity (FN) and its amplitude was gradually increased with an increasing number of preceding distractor episodes. This ERP effect has been related to a frontal suppression process [17, 20]. A process which in turn might be associated with corresponding changes in the subsequent processing of the cue-target compound: In case of successful target detection, the ERPs

are characterized by a prominent P3 amplitude, which is significantly diminished in case of a miss [18]. In visual DIB, earlier ERP components related to sensory processing were not affected by the preceding distractors. Following the authors [17, 20], the FN should signal the activation of a negative attentional set which does not suppress the sensory (here: visual) analysis of the target, but which prevents updating processes of stimuli characterized by distractor, respectively, target features in working memory.

However, the predictive values of the P3 response for target detection might be questioned if the target is defined in the auditory modality: For unimodal DID (cue and target presented in the auditory domain), ERP responses to the cue-target compound were still characterized by a prominent P3 –if the target is detected successfully [23]. Yet, correctly reported targets were additionally characterized by a stronger fronto-central negativity at about 200 ms. What is more, for the cross-modal DID effect (visual cue and auditory target), successful target detection was exclusively characterized by a pronounced fronto-central negativity, whereas the P3 amplitude was not enhanced when contrasted to misses [24]. The authors proposed that the process signaled by the fronto-central negativity might be related to a processing negativity [29]. Since this ERP signature shares the characteristics of an auditory awareness negativity (AAN) [30, 31], target access might also be related to a modality-specific, and not to a post-perceptual, process [26]. In sum, these deviant ERP results signal that the central suppression process previously identified in unimodal visual distractor studies must be questioned, if cue and target are defined in different modalities. The processing of a target stimulus in a multiple-distractor task seems to be affected by the congruency of the cue's modality.

To this end, the current study was set out to explore the mechanism of distractor processing and its effect on the processing of the target in a cross-modal DIB setup. Distractors as well as the target were defined in the visual domain. In line with previous unimodal DIB studies [12, 13, 15, 16, 18–20], we used a dynamic random dot pattern, and distractor as well as target episodes were defined by a change of the local elements (here: shape). The occurrence of the target was signaled by an auditory cue. A previous behavioral study already indicated that target detection will depend on the number of distractors presented in the pre-cue epoch [22]. More specifically, we stated the following research questions:

1. **Can we replicate the behavioral cross-modal DIB effect?**
   Following the findings of Kern and Niedeggen [22], the key characteristic of cross-modal DIB is expected to be replicated: Target detection is predicted to depend on the number of preceding distractor episodes [13]. This functional relationship will be mostly expressed if cue and target are presented simultaneously [12].

2. **Which ERP components can be related to the processing of distractors, and can we identify a cumulative process?**
   Based on previous ERP results, we assume that the central inhibition will be driven by the visual distractors. Given that a cumulative distractor effect can be identified in detection performances (see question 1), we assume that the distractor-evoked potentials will share the characteristics of the ERP described in previous DIB studies: The gradual activation of the inhibition process should reveal itself in an increasing FN [17, 20, 26, 28].

3. **Which ERP components characterize the processing of the cue-target complex during cross-modal DIB? And how are those ERP signatures related to the preceding distractors?**
   In line with our predictions regarding the neural response to distractors (see [2]), we furthermore assume that the processing of the visual target in the DIB setup is primarily determined by the presence of the, likewise, visual distractors–but does not critically depend on

the type of cue, as previously assumed by unimodal visual research [14]. If the cue merely serves as a temporal marker to deactivate the negative attentional set, its modality should not affect cue-target ERPs. Hence, ERP signatures of target processing and access should be consistent with the results from previous unimodal DIB experiments [17, 18, 20]: The occurrence of the target should elicit a distinct P3 component. We furthermore assume that its amplitude should increase in experimental conditions where the probability for target detection is high–this will be the case if the number of preceding distractor episodes is low [13].

## 2. Materials and methods

The PsychoPy code, along with all data collected and stimuli utilized in this experiment, are made available in an open repository (https://doi.org/10.17632/93nxgxntbk.1).

### 2.1. Participants

The sample size, needed to detect the crucial cumulative distractor effect in the ERPs reliably in the experiment, was a priori calculated with the G*Power software [32] version 3.1 and a significance level ($\alpha$) of .05, as well as a desired power of 80% for an F-test with repeated measurements. Following Niedeggen et al. [20], we anticipated a medium effect size (f = .25) for the within-subject factor 'number of distractors' (1 vs. 3–4 vs. 5–6) regarding the increasing FN of distractor-evoked ERPs. This calculation indicated a total sample size of N = 28 as necessary for the experiment.

Participants were recruited from the university environment of Freie Universität Berlin between 1st of May and 7th of August 2023. They were either given course credit or a compensation of 10€/h for their participation. The experimental procedure described in the following was approved beforehand by the local ethics committee at Freie Universität Berlin (027/2019).

All participants provided written informed consent before their participation and furthermore confirmed that they had no history of neurological or psychiatric conditions, had normal or at least corrected-to-normal vision, had unimpeded hearing abilities, and had no history of substance abuse. Moreover, individuals with contraindications concerning EEG application (e.g., scalp irritation) were not considered eligible to participate. In total, 34 participants took part in the experiment.

The exclusion of datasets was based on a set of criteria already established in previous ERP studies [23, 24] and necessary to ensure the reliability of our findings. These criteria included the following: First of all, [1] insufficient target detection in the zero-distractor condition (less than 60% accuracy). It is highly probable that individuals showing strongly impaired target detection in the zero-distractor condition may not have understood the task correctly or are unable to perceive the stimuli adequately, which would introduce confounding factors to our experiment. Another exclusion criterion [2] was based on the number of electrophysiological artefacts (eye blinks, head movements, or high electrophysiological alpha activity (>80 μV)): Participants' data were excluded if the analyses was based on less than 10 valid trials in a condition. Due to the first exclusion criterion, no dataset had to be discarded. Due to the second criterion, the datasets of seven participants had to be discarded. The final sample consisted of 27 participants (23 women; 18–34 years of age; $M_{age}$ = 22.30, SD = 3.770).

### 2.2. Stimuli, procedure, design

The experiment took place in a sound-dampening chamber. Participants had their head stabilized on a chin rest and were seated in a distance of 62 cm from a 20-inch Sony Trinitron

Multiscan G520 monitor with a resolution of 1280 x 1024. As the only light source in the room, subdued, indirect lighting was implemented from the ceiling to avoid reflections on the computer screen. Participants were equipped with Audio-Technica ATH-LS70iS in-ear head-phones, each fitted with individual earpieces and a customizable ear mount. The experiment was executed on a Windows PC, utilizing PsychoPy software (Version 3.6.8 for Windows).

The stimuli used in the present experiment were not created in real-time. We generated predefined audio-visual stimulus sequences using a custom program written in Python (v.3.6) and PsychoPy (v.2022.1.1, [33]) to ensure precise timing alignment between the visual and auditory components. As a result, the auditory stimulation consistently commenced simultaneously with the visual stimulation, without temporal jitter.

In the visual domain (see Fig 1), participants were exposed to a centered fixation point defined by a single static white dot (0.46˚ in diameter; HSV: [0 0 100]). The fixation point was centered on a light-grey circular patch (3.23˚ in diameter; HSV: [0 0 50]) in the middle of the screen. The light-grey patch was enclosed by a larger dark grey patch (18.77˚ in diameter; HSV: [0 0 40]) including the local elements defining the RSVP stream. Within this stream, there were two sets of each 100 uniformly moving dots (0.69˚ in diameter). One dot pattern was homogeneously colored in dark blue (HSV: [240 100 100]) and moved in a clockwise motion in the foreground (0.3˚ per frame), while the other pattern was homogeneously colored in bright green (HSV: [120 100 100]) and moved counterclockwise in the background (0.3˚ per frame). Hence, blue dots could temporarily cover the green dots at times. While the blue dots retained their circular shape for the entire trial duration (5,000 ms), the shape of the green elements could transiently change from a circular shape (dots) to a rectangular shape (squares), and back. A transient change of shape of the green pattern (dot to square for 100 ms) indicated a distractor event, when it appeared before the cue (see below), and a target event when it appeared simultaneously (SOA = 0 ms) or shortly after the cue (SOA = 300 ms). Fig 1 illustrates this experimental setup as well as the sensory stimulation of the experiment. Please note that a similar setup has been used in previous unimodal DIB studies [21]: Here, the phasic changes of the local elements were defined by color changes.

In addition to the RSVP stream, two rapid serial auditory presentations (RSAPs; each 5,000 ms in duration) were played simultaneously to the participants via headphones (see Fig 1). The auditory stimuli of both streams in the present study were developed using the "Tone Generator" and "WavePad Editor" programs from NCH Software (Greenwood, USA) and were already used in previous experiments [22–24]. One RSAP stream presented to the left ear consisted of a continuous sinusoidal tone defined by a 5 Hz modulation within a frequency range of 270 to 330 Hz. As compared to the other channel, amplitude was slightly reduced (-20 dB). This stream provided the cue event which was defined by a 100 ms increase in amplitude (+10 dB). Participants were explicitly directed to monitor whether the cue appeared in a given trial or not. The cue indicated to shift the participant's attention to the visual stream and to further verify the presence of the target stimulus. The second RSAP stream, played to the right ear, comprised 50 pure sine-wave tones, each lasting 30 ms with a 70 ms inter-stimulus interval. These tones were randomly selected from a set of seven, falling within the frequency range of 1,800 to 2,200 Hz. The second stream was not task relevant–but previous DID experiments revealed that the concurrent stream prevents an automatic detection of the cue (amplitude increase) [23, 24].

In each experimental trial, the RSVP and RSAP streams (5,000 ms) were combined and followed by two alternative-forced-choice questions: Participants were required to, firstly, determine if they perceived an auditory cue (question 1: "Did you hear a change in the continuous tone?") and, secondly, if they observed a visual target accompanying or following the cue shortly after (question 2: "Did one of the patterns change simultaneously with or shortly after a

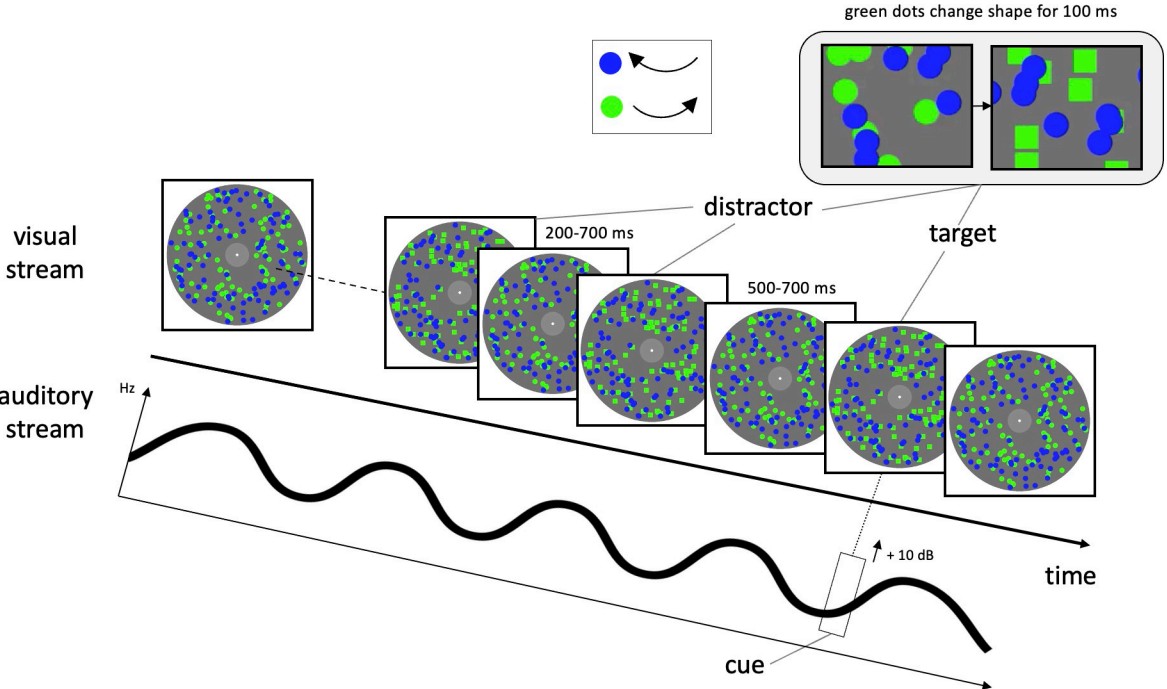

**Fig 1. Design of an experimental trial from the multiple distractor condition.** Depicted is the experimental design of a trial from the multiple distractor condition. While the cue was specified within the auditory modality, target and distractors were presented in an RSVP stream. The only factor distinguishing target from distractor stimuli was the variation in timing of their presentation: distractors were presented before the cue onset, whereas a target could coincide with (SOA = 0 ms, as illustrated here) or shortly follow the cue (SOA = 300 ms). Visual stimuli were incorporated into an RDK, characterized by two overlapping motion planes moving in opposite directions. On the right ear, participants were exposed to a series of pure tones, while a continuous tone was played to the left ear. The cue was identified as a brief increase in amplitude (+10 dB) in the continuous tone. The visual target event was marked by the alteration of green dots, transitioning from a circular to square shape (for 100 ms). RSVP = rapid serial visual presentation, SOA = stimulus-onset asynchrony; RDK = random dot kinematogram.

change in the tone?"). Responses (yes/no) were indicated via button press on a keyboard, without any time limit or rush. However, participants were instructed to provide responses with maximum accuracy.

The experimental designs included two experimental factors: If a cue and a target were presented, the temporal delay to the cue (cue-target SOA) was systematically varied. In 70 trials each, cue and target were presented simultaneously (SOA 0 ms), or with a delay of 300 ms (SOA 300 ms). As a second factor, the number of visual distractors preceding the cue and target was systematically varied (0 vs. 1 vs. 3–4 vs. 6–8 distractors) in each participant. Within each of the two SOA conditions, the distractor conditions "1," "3–4," and "5–6" were each represented in 20 trials, and the "0 distractor" condition was represented in 10 trials. This number of trials was based on the outcome of a pilot study (n = 15 participants), see supplement (S1 Table). Most importantly, the pilot data confirm the reliability of the effect of distractor number.

To assess the reliability of participant's responses in terms of false alarms (falsely reported targets after correctly detected cues), 50 control trials solely contained the cue (15 trials for each of the three distractor conditions containing 1, 3–4, or 5–6 distractors, as well as 5 trials for the condition without any distractor). As an additional control condition, 50 trials included neither the cue nor the target (again 15 trials for each of the three conditions containing 1, 3–4, or 5–6 distractors, as well as 5 trials for the condition without any distractors). Hence, it was possible to control for a potential response bias in participants. As mentioned above, the

setup was tested in a behavioral pilot study (n = 15 participants), and data revealed a low probability in false alarm rates as well as a significant distractor effect.

For the analysis of the distractor-evoked potentials, before artefact rejection, 55 trials were available for each distractor condition (1 vs. 3–4 vs. 5–6), with the SOA and control conditions combined. In previous studies, the analysis of distractor-evoked ERPs was based on a comparable number (n = 63 trials per distractor condition; [17, 21]). ERP responses recorded in the trials containing no distractor were not further considered in the analysis. Experimental and control conditions added up to a number of n = 240 trials in total.

To probe the brain's response to distractors, we specified, in each trial containing one or multiple distractors, a single distractor which was temporally segregated from other events to avoid superimposition effects in the EEG signal. Always the final distractor of a sequence served as probe. The onset of the upcoming cue after the probe randomly varied between 500 and 700 ms. The onset of a preceding distractor was at least 500 ms before the onset of the probe. In trials where only one distractor was present, the distractor stimulus was temporally synchronized with the final distractor from a sequence of multiple distractors (500–700 ms before the cue onset). These time windows have been established in previous ERP studies [17, 20, 21, 26]. They furthermore ensure that participants do not confuse the last distractor and target epochs [13]. In summary, there was no distractor, or any other stimulus, present from 500 ms before to 500 ms after the onset of the probe. SOAs between successive distractors ranged from 200 to 700 ms.

The general experimental procedure foresaw the following steps: Once participants provided informed consent, the testing commenced with a pretest comprising 32 practice trials, intended to acquaint participants with the stimuli and tasks. After each trial, the experimenter gave verbal feedback regarding accuracy of the participants' responses. If participants showed a consistent target detection (>60%), the EEG set was mounted, and the main experiment was started.

The main experiment consisted of 240 trials in total, presented in a randomized order for each participant, and lasted approximately 45 minutes. Participants were allowed and encouraged to take short breaks by withholding their response.

Behavioral results were compared to the results of a previous cross-modal DIB study [22]. Although a distractor effect was observed in the previous study (see above), differences in the visual setup must be considered: In the study of Kern & Niedeggen [22], visual distractor and target stimuli were defined as a local luminance or color change embedded in the simple clockwise motion of a preloader signal. In the current study, a more complex dynamic pattern was used (two superimposed motion planes). This setup has been previously established by Valdes-Sosa and colleagues [34–36] and requires a visual segregation process. We will refer to this difference in visual stimuli configuration as 'complexity' in the following.

A second difference lies in the definition of the factor 'distractor number': Kern and Niedeggen [22] defined three classes (0, 1, and 6–8 distractors), whereas the current study used a more equidistant classification (0, 1, 3–4, and 5–6 distractors). To allow a statistical comparison, the conditions '5–6 distractors' in the recent study and '6–8 distractors' in the previous study were both defined as an 'n distractor' condition. To derive a cumulative effect of distractor number, this condition was used for the comparison.

The data from the earlier study are publicly available in an open repository (https://doi.org/10.17632/wxmhwv7xvd.1).

## 2.3. EEG recording and analysis

To initiate the preparation of the EEG recording, an elastic cap (EASYCAP, Herrsching-Breitbrunn, Germany), featuring predetermined electrode positions based on the 10-20-system

[37], was mounted onto the participant's head (equidistant montage file). Twenty-nine active Ag/AgCl electrodes were referenced to linked earlobes, while the ground electrode was located at the FCz position. Impedance levels across all electrodes were maintained below 10 kΩ. An electrooculogram monitored vertical and horizontal eye movements through electrodes at the outer canthi (hEOG) and along the sub- and supraorbital ridges of the right eye (vEOG). EEG signals were captured through a 40-channel NuAmps amplifier (Software Acquire, Version 4.3, Neuroscan Labs, Neurosoft Inc., El Paso, TX, USA). The data were online band-pass filtered within the frequency range of 0.1–100 Hz and were sampled at a rate of 500 Hz.

Subsequently, the data were analyzed offline using the "BrainVision Analyzer" software (Version 2.1, Brain Products GmbH, Gilching, Germany). To identify the relevant ERP components, the EEG data of each participant was segmented into 900 ms epochs, covering a time range from -100 to 800 ms in reference to two distinct markers: The first class of EEG segments was referenced to the onset of the final distractor in a sequence of multiple distractors or the solitary distractor in the single-distractor condition (regarding analysis of distractor-evoked ERPs). This class of segments served as a probe for the distractor processing (see research question 1). The second class of EGG segments was referenced to the onset of the cue-target compound in the SOA 0 ms condition. With respect to research question 2, these signals served as a probe of target access. Subsequently, these segments underwent filtering (0.3–30 Hz, with a 50 Hz Notch filter) and were baseline corrected (-100 to 0 ms). A semi-automatic artifact rejection procedure was employed to identify and exclude isolated EEG data points containing ocular or muscular artifacts, slow drifts caused by (head) movements, or pronounced EEG alpha activity ($> 80$ µV) from further analysis.

Per participant, the electrophysiological analysis of the probe state was finally based on a mean of 27.15 (SD = 6.73) out of 55 trials from the 'single distractor' condition, 27.00 (SD = 6.55) out of 55 trials from the '3–4 distractors' condition, and 25.19 (SD = 7.55) out of 55 trials from the '5–6 distractors' condition, after the artefact rejection process. The analysis of the cue-target access was based on a mean of 45.07 (SD = 7.35) out of 70 trials from the 'cue-target compound', as well as 30.44 (SD = 4.76) out of 50 trials from the 'cue only' condition. The rejection rate in the present study was comparable to that of a previous experiment ([24]: In the distractor-sensitive subgroup, a mean of 50.76 hit trials (SD = 12.19) out of 80 trials remained after artifact rejection; In the distractor-insensitive subgroup, a mean of 38.77 hit trials (SD = 9.97) out of 80 trials remained after artifact rejection), leading to an analogous signal-to-noise ratio of the averaged ERPs in single participants. Nevertheless, to test the reliability of our data, we conducted an additional analysis of the distractor ERP, where a loss of trials after artefact rejection was particularly pronounced. For this additional analysis we implemented a more liberal, fully automated rejection process, controlling exclusively for eye blinks (vEOG channels >80 µV), transient over regulations (no signal for more than 100 ms), and alpha activity in the baseline range of the frontal electrodes (+40 to -40 µV between -100 and +100 ms). Artefacts at other electrodes and drifts–usually controlled for manually—were not considered. The results of this additional analysis replicated our findings and can be found in the supplement (see S2 Table and S1 Fig). The additional dataset is available in the open repository (https://doi.org/10.17632/93nxgxntbk.1).

All remaining EEG signal sweeps were averaged by analysis focus: While all segments were separated for electrode position, the first class of segments (probe state) was additionally separated according to the number of distractors (1 vs. 3–4 vs. 5–6). The second class of segments (cue-target access) was additionally separated according to the target presence (cue-target vs. cue only). The 'cue-target compound' was furthermore separated regarding the number of preceding distractors (high vs. low).

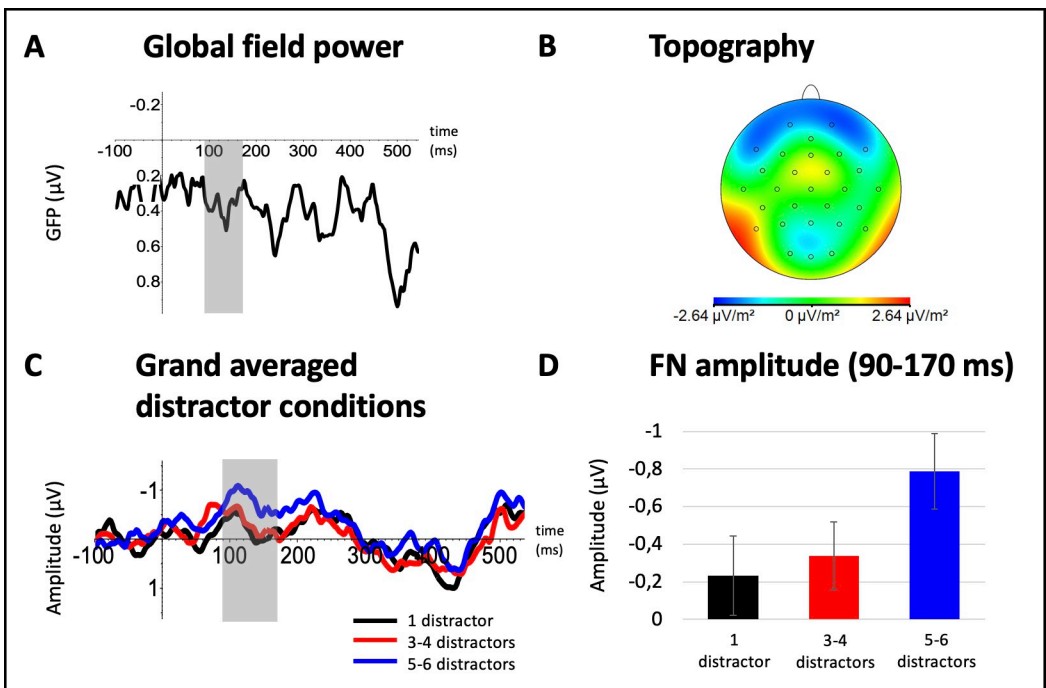

**Fig 2. Global field power, topographical mapping, and grand averages as well as mean amplitudes of the distractor-evoked ERP (FN, 90-170ms).** (A) Global field power plot showing the root mean squared power (in μV) for the difference wave between the '1 distractor' and '5–6 distractors' condition. A peak can be identified at 130 ms. The symmetrical analysis window for the a priori defined ERP component of interest, FN (90–170 ms), is shaded in grey. (B) Topography of the frontal negativity (FN) based on reference-independent maps of the difference wave between the '1 distractor' and '5–6 distractors' condition. An anterior maximum for the FN (90–170 ms) can be identified. (C) For the anterior cluster, grand averages (GAs) are presented for trials containing only 1 distractor (black), for trials containing 3–4 distractors (red), and for trials containing 5–6 distractors (blue). The analysis window for the a priori defined ERP component of interest, FN (90–170 ms), is shaded in grey. (D) A significant linear trend could be identified for the FN, revealing a gradual increase of the component's negative mean amplitude (in μV) with an increasing number of distractors.

As for the ERPs related to the onset of the final distractor (in the following labeled as distractor-related ERPs), we aimed to identify a gradually increasing frontal negativity in the ERP responses evoked by visual distractors. Previous studies have reported that the latency of the FN might depend on the visual feature chosen: For motion, the signal appears at about 225–275 ms [26]. For orientation, at about 300–500 ms [20] or 250–450 ms [17]. For color, Winther & Niedeggen [21] reported 325–475 ms. This variability–probably also affected by other differences in the experimental manipulation–prevented that a time range could be identified a priori. Instead, we employed visual inspection of the grand averaged ERP signals at frontal leads. To confirm the visual inspection of a suitable time window, the global field power (GFP) of the difference wave between the '1 distractor' and '5–6 distractors' condition was computed (see Fig 2A). Based on the local maximum of the GFP (120 ms), a symmetrical time window ranging from 90–170 ms was determined. Following previous studies [17, 20, 21], an anterior cluster of electrodes was defined covering the FN (AFz, F7, F8, Fp1, Fp2).

Regarding our hypothesis concerning the cue-target-evoked ERP signatures, we aimed to test the replicability of the P3 as indicator for visual target access. To this end, the temporal progression of global activity, defined by the GFP, and its connection to the spatial distribution of electrical activity induced by the processing of the cue-target compound were examined. A slow parietal ERP positivity, starting at about 350 ms and lasting for about 150 ms, could be identified through visual inspection of grand averages. Correspondingly, the GFP of the

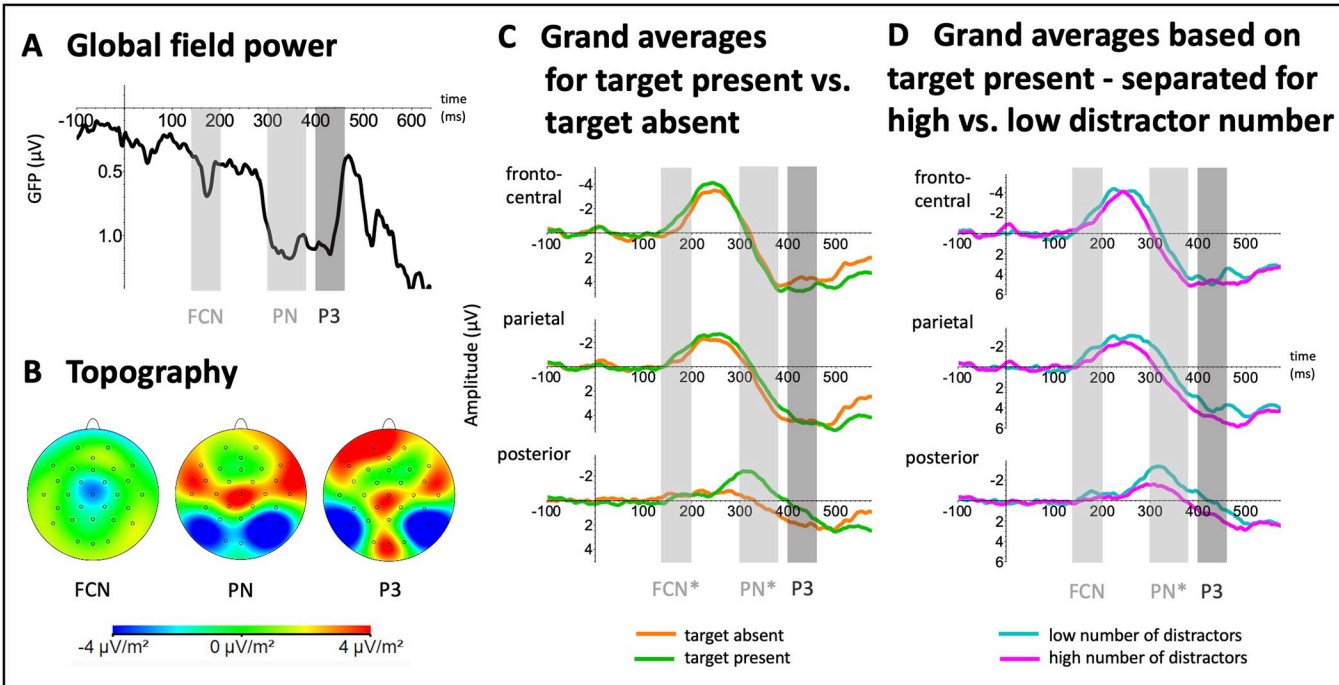

**Fig 3. Global field power and grand averages for the cue-target-evoked ERPs as well as their respective topographical mapping.** (A) Global field power plot showing the root mean squared power (in µV) for the difference wave between the 'cue-target' and 'cue-only' condition. A first peak in activation can be identified at 170 ms (FCN), followed by another peak at 340 ms (PN). Finally, a peak at 430 ms (P3) becomes visible. The symmetrical analysis window for the a priori defined ERP component of interest, P3 (400–460 ms), is shaded in darker grey, while the windows for the a posteriori defined FCN (140–200 ms) and PN (300–380 ms) are indicated in light grey. (B) Topography of the fronto-central negativity (FCN), the posterior negativity (PN), and the P3 based on reference-independent maps of the difference wave between the 'cue-target and 'cue-only' condition. A broad anterior topography for the FCN (140–200 ms) can be identified, as well as a locally restricted posterior maximum for the PN (300–380 ms) and a parietal topography for the P3 (400–460 ms). (C) Grand averages for trials when the target was absent (orange) as well as when it was present (green) are presented separately for the fronto-central, parietal, and posterior electrode cluster. The analysis window for the a priori defined ERP component of interest, P3 (400–460 ms), is shaded in darker grey, while the windows for the a posteriori defined FCN (140–200 ms) and PN (300–380 ms) are indicated in light grey. While no significant difference in the P3 between could be found, the presence of the target was associated with a significantly stronger expression of the FCN and PN (indicated by *). (D) Grand averages for trials containing a low number of distractors (cyan), associated with a higher target detection probability, as well as trials containing a high number of distractors (purple), associated with a lower target detection probability are depicted separately for the fronto-central, parietal, and posterior electrode cluster. The analysis window for the a priori defined ERP component of interest, P3 (400–460 ms), is shaded in darker grey, while the windows for the a posteriori defined FCN (140–200 ms) and PN (300–380 ms) are indicated in light grey. The PN is the only component for which a significant difference (indicated by *) can be observed between the two conditions, demonstrating a higher amplitude for an increased detection probability.

difference wave between the 'target present' and 'target absent' condition shows a local maximum at 430 ms (see Fig 3A). In coherence to the definition of the FN, described above, a symmetrical time window ranging from 400–460 ms (further labeled as P3) was chosen.

Additionally, the post-hoc analysis of the grand-averaged potentials revealed two further components: An early peak at fronto-central electrode leads, starting at 140 ms and lasting for about 150 ms, shared the temporal and spatial characteristics of the early negativity reported in previous uni- [23] and cross-modal DID studies [24]. It was furthermore recognized in the GFP with a corresponding peak at 170 ms (see Fig 3A), therefore a symmetrical analysis window was chosen ranging from 140 to 200 ms around the GFP peak.

Second, an occipital negativity, starting at 280 and lasting for about 100 ms, was identified in the grand-averaged ERP data. This signature corresponded to a peak in the GFP of the difference wave between the 'target present' and 'target absent' condition at 340 ms (see Fig 3A). The analysis window was set symmetrical around this peak, ranging from 300 to 380 ms. Although not expected a priori, we included these two components in an a posteriori analysis.

Previous research [20, 23] provided a template for sensitive electrode clusters regarding cue-target-evoked ERPs. Inspection of the reference-independent topographical distribution (Fig 3B) led to a minor adaption of the three clusters (fronto-central: FC1, FC2, Cz, CP1, CP2; parietal: P3, P4, Pz, CP1, CP2; posterior: O1, O2, Oz, P7, P8). All cue-target-evoked components were contrasted regarding their expression according to whether the target was present (cue-target compound) or absent (cue only condition), to elucidate whether they are associated with the processing of the visual target stimulus.

## 2.4. Statistics

For data analysis, IBM SPSS Statistics (version 28) was used. In all analyses of variance, degrees of freedom were corrected according to the Greenhouse-Geisser criterion [38], if the assumption of sphericity was violated.

**2.4.1. Behavioral analysis.** To analyze the behavioral data, the target detection rate of each participant, referring to a correctly detected target after a correctly detected cue, was calculated for each experimental condition separately. This was followed by a descriptive analysis and a two-way repeated measures analysis of variance (rmANOVA) containing the within-subject factors 'number of distractors' (0 vs. 1 vs. 3–4 vs. 5–6) and 'SOA' (0 vs. 300 ms). In case of a significant interaction, pairwise comparison was performed as post hoc analysis.

Furthermore, our data were compared with the behavioral results of a previous cross-modal DIB study [22]. In total, this earlier study comprised three experiments: While in one experiment, an individual deviant tone was implemented as the cue (see experiment 2; 22), the other two experiments employed the same rise in amplitude of a continuous tone as we (see experiment 1 & 3; 22). In comparison to our stimulus setup, the latter two experiments differed solely in the visual feature defining target and distractors (local luminance change vs. color change) and were hence selected for the subsequent analysis. To improve statistical power, both of these earlier datasets were merged together and handled as a single set. This action was feasible due to the absence of any significant difference concerning the effect of the target feature (see: 22). An rmANOVA was conducted with the within-subject factor 'number of distractors' (0 vs. 1 vs. n) and the between-subject factor 'target feature' (complex vs. simple), where 'complex' refers to our stimuli arrangement and 'simple' to Kern and Niedeggen's [22] two stimulus configurations. In case of a significant interaction, pairwise comparison was performed as post hoc analysis. We furthermore conducted an a priori trend analysis according to orthogonal polynomial contrast coefficients.

**2.4.2. ERP analysis.** In line with our EEG-related research objectives, the analysis of ERPs focused on two aspects: distractor-evoked as well as cue-target-evoked components. Regarding the former, mean amplitudes were extracted in the time range of the early FN (90–170 ms) from the anterior electrode cluster. An rmANOVA for the within-subject factor 'number of distractors' (1 vs. 3–4 vs. 5–6) as well as the within-subject factor anterior 'electrode' (AFz vs. F7 vs. F8 vs. Fp1 vs. Fp2) was calculated. The latter factor was included to test the homogeneity of the electrode cluster defined with respect to the effect of experimental manipulation. Therefore, an interaction effect between the two within-subject factors was of primary interest. This applies to all rmANOVAs related to ERPs that we have conducted. However, for the FN specifically, as described already in previous studies [17, 20, 21], a trend analysis according to orthogonal polynomial contrast coefficients was performed to identify a functional relationship between the FN's amplitude and the number of preceding distractors.

For the analysis of the cue-target-evoked ERP signatures, mean amplitudes were extracted in the time range of the P3 (400–460 ms), and exploratory also of the early fronto-central negativity (140–200 ms), and the posterior negativity (300–380 ms). In a first step of analysis, the

contribution of the ERP component to target processing was identified. To this end, the conditions 'cue-only' (target absent) and 'cue-target' (target present) were compared. At first, P3 amplitude was compared by running a rmANOVA, including the within-subject factor parietal 'electrode' (P3 vs. P4 vs. Pz vs. CP1 vs. CP2). As the topographically widespread fronto-central negativity in our data resembled an early frontal process already identified in previous cross-modal research [23, 24], we focused our analysis of this ERP signature on the fronto-central cluster. Another rmANOVA for the within-subject factor 'electrode' (FC1 vs. FC2 vs. Cz vs. CP1 vs. CP2) as well as 'target presence' (cue only vs. cue-target) was computed. Finally, for the temporo-occipital negativity, an rmANOVA for the within-subject factor 'electrode' (O1 vs. O2 vs. Oz vs- P7 vs. P8) of the posterior cluster as well as 'target presence' (cue only vs. cue-target) was calculated.

In case of a significant difference, the contribution of the ERP component to successful target detection was further examined. To this end, we averaged the ERPs preceded by a low number of distractors (0, 1) and the ERPs preceded by a high number of distractors (3–4, 5–6). The corresponding conditions differed significantly with respect to target access (see behavioral results), and differences in the ERPs can therefore be attributed to the detection performance. Consequently, we ran rmANOVAs for the within-subject factor 'number of distractors' (high vs. low), as well as 'electrode' of the determined cluster. Please note, that a direct comparison of the conditions 'target-hit' vs 'target-miss'–as applied in previous studies [see 20, 23, 24]—was not possible because the number of misses was not sufficient for an averaging per participant (on average only 29 misses per subject). Due to the concurrent evaluation of distractor-evoked as well as cue-target-evoked potentials in this study, the study design did not permit the planning of sufficient trials to obtain an adequate number of misses (see limitations).

To sum up, the components were furthermore contrasted regarding their expression according to a high versus low probability of target detection. With a significant distractor effect present in the behavioral data (see behavioral results), it could be inferred that conditions containing none or only a low number of distractors are associated with a higher probability of target detection, whereas conditions containing multiple distractors (3–4 or 5–6) are associated with a lower probability of target detection.

## 3. Results

### 3.1. Behavioral results

Mean target detection rates for all experimental conditions are summarized in Table 1.

**Table 1. Mean target detection rates for the experimental conditions.**

| Distractors | SOA 0 ms | SOA 300 ms |
|---|---|---|
| 0 | M = 94.74 | M = 93.46 |
| | CI [91.05, 98.44] | CI [89.40, 97,52] |
| 1 | M = 88.36 | M = 87.00 |
| | CI [83.41, 93.31] | CI [79.16, 94.85] |
| 3–4 | M = 82.37 | M = 89,78 |
| | CI [75.64, 89,10] | CI [84.01, 95,46] |
| 5–6 | M = 74,37 | M = 86.25 |
| | CI [66.15, 82.59] | CI [78.70, 93.79] |

Mean (in %) and the corresponding 95% confidence interval (CI) for averaged correct target detection after correct cue detection, reported separately for each distractor and SOA condition. SOA = stimulus onset asynchrony.

The rmANOVA yielded a significant difference in the detection rate between the SOA conditions, with a higher detection rate at SOA 300 as compared to SOA 0 (factor 'SOA': $F_{(1, 26)}$ = 5.36, p < .029, $n_p^2$ = .171). Data analysis showed that the number of distractors significantly influenced target detection: The baseline detection rate (0 distractors: 95%) was gradually reduced to 74% in the condition with the highest number of distractors, for an SOA of 0 ms. For an SOA of 300 ms, the baseline detection rate (0 distractors: 93%) decreased to 86% in the highest distractor condition. This resulted in a significant main effect of the factor 'distractor number' ($F_{(1.87, 48.51)}$ = 17.31, p < .001, $n_p^2$ = .400). The effect of distractor number was more expressed at SOA 0 as compared to SOA 300, indicated by the significant interaction of both factors ('distractor number' x 'SOA': $F_{(3, 78)}$ = 8.59, p < .001, $n_p^2$ = .248). Furthermore, a significantly different linear trend emerged in the means of the detection rate with respect to the levels of both factors ($F_{(1, 26)}$ = 18.65, p < .001, $n_p^2$ = .418).

In order to determine in which SOA condition (0 ms vs. 300 ms) the factor 'distractor number' had a greater impact on target detection, analysis was run separately for the SOA conditions: For an SOA of 0 ms, the detection rate dropped from 95% (0 distractors), to 88% (1 distractor), to 82% (3–4 distractors), to eventually 74% (5–6 distractors) (see Table 1). The effect of the number of distractors was strongly expressed ($F_{(2.03, 52.89)}$ = 20.92, p < .001, $n_p^2$ = .446) and followed a linear trend ($F_{(1, 26)}$ = 32.27, p < .001, $n_p^2$ = .554). Post hoc pair-wise comparisons revealed that hit rates decreased with an increasing number of distracting stimuli for each gradation (see Table 2).

For the long SOA, the drop in detection rate was smaller, from 93% (0 distractors), to 87% (1 distractor), to 90% (3–4 distractors), to finally 86% (5–6 distractors) (see Table 1). Nevertheless, the effect of 'distractor number' (SOA 300 ms: $F_{(3, 78)}$ = 4.03, p = .010, $n_p^2$ = .134) was also significantly expressed, albeit more weakly than in the small SOA condition. Less pronounced was furthermore the significantly linear trend ($F_{(1, 26)}$ = 5.24, p = .030, $n_p^2$ = .168).

**Table 2. Summarized statistical results of the post hoc pair-wise comparisons between the hit rates of the different distractor conditions, separated for each SOA condition.**

| | SOA condition | |
|---|---|---|
| **Comparison** | **0 ms** | **300 ms** |
| 0 vs. 1 distractors | **$F_{(1, 26)}$ = 11.10**<br>**p = .003**<br>**$n_p^2$ = .299** | **$F_{(1, 26)}$ = 6.51**<br>**p = .017**<br>**$n_p^2$ = .200** |
| 0 vs. 3–4 distractors | **$F_{(1, 26)}$ = 18.09**<br>**p < .001**<br>**$n_p^2$ = .410** | $F_{(1, 26)}$ = 2.41<br>p = .133<br>$n_p^2$ = .085 |
| 0 vs. 5–6 distractors | **$F_{(1, 26)}$ = 31.42**<br>**p < .001**<br>**$n_p^2$ = .547** | **$F_{(1, 26)}$ = 7.97**<br>**p = .009**<br>**$n_p^2$ = .235** |
| 1 vs. 3–4 distractors | **$F_{(1, 26)}$ = 8.60**<br>**p = .007**<br>**$n_p^2$ = .249** | $F_{(1, 26)}$ = 1.22<br>p = .280<br>$n_p^2$ = .045 |
| 1 vs. 5–6 distractors | **$F_{(1, 26)}$ = 27.13**<br>**p < .001**<br>**$n_p^2$ = .511** | $F_{(1, 26)}$ = .134<br>p = .717<br>$n_p^2$ = .005 |
| 3–4 vs. 5–6 distractors | **$F_{(1, 26)}$ = 9.84**<br>**p = .004**<br>**$n_p^2$ = .275** | $F_{(1, 26)}$ = 4.86<br>p = .037<br>$n_p^2$ = .157 |

Post hoc pair-wise comparisons were conducted between the hit rates of all four distractor conditions (0 vs. 1 vs. 3–4 vs. 5–6 distractors), resulting in six analyses per SOA condition (0 vs. 300 ms). In case of a significant effect, the result is printed in bold. SOA = stimulus onset asynchrony.

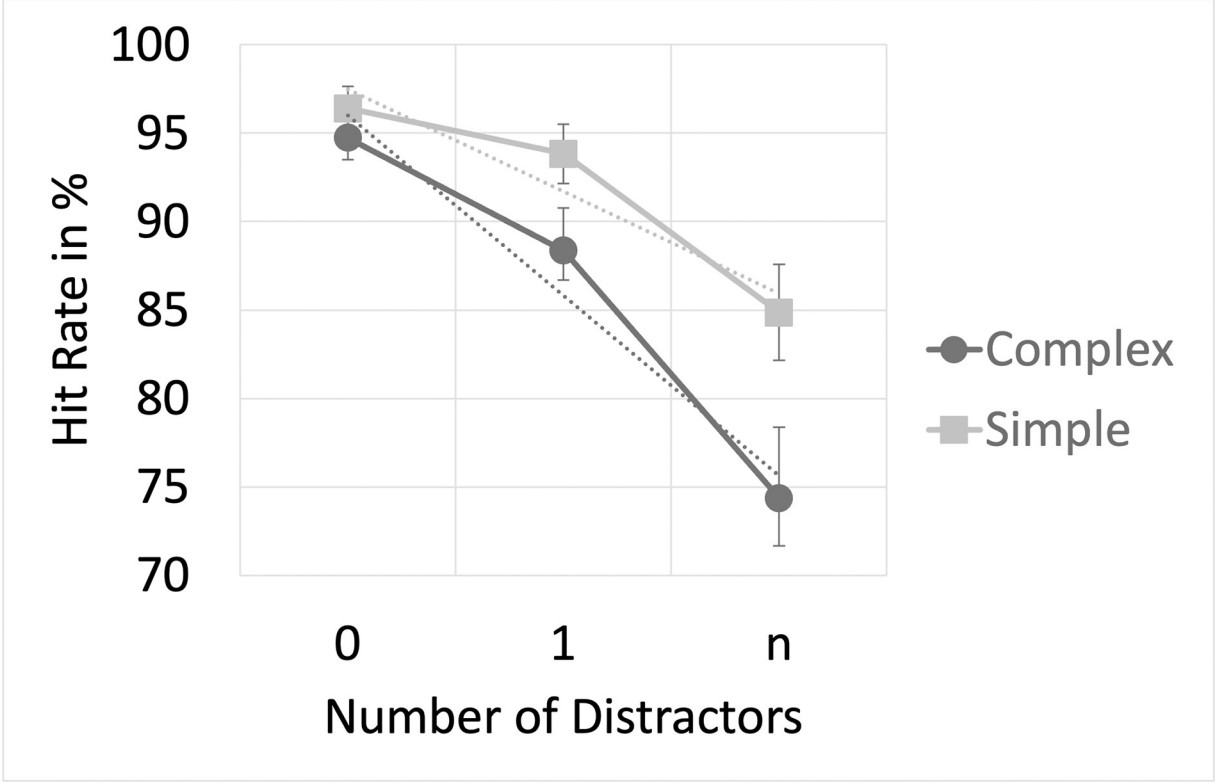

**Fig 4. Distractor effects at cue-target SOA 0 ms for different complexity levels of the visual stimulus configuration.** Depicted are mean target detection rates (hit rates), and their corresponding standard error, acquired in the present experiment ('complex' condition) in comparison to hit rates from two summarized previous cross-modal DIB experiments ('simple' condition) at cue-target SOA 0 ms. The experiments solely varied in their visual stimulation (target/distractors). The auditory stimulation (cue) as well as the experimental procedure, and the overall cross-modal setup (auditory cue, visual target/distractors) was identical. While our experiment utilized a change of shape as target/distractor events implemented in a complex stimulus display, the pooled previous experiments presented either a change of luminance or color embedded in a rather simple stimulus configuration. Overall, the reduction of hit rates with an increasing number of distractors was more pronounced for the complex setup. A significant interaction exists between the distractor number and the utilized type of visual display, indicating that the magnitude of the cross-modal distractor effect is related to the complexity of the visual target's/distractor's presentation mode. SOA = stimulus onset asynchrony.

Post hoc pair-wise comparisons showed that, in particular, the condition with 3–4 distractors did not contribute as much to a significant effect of the factor 'distractor number' as the other conditions (see Table 2).

The examination of control conditions without a target demonstrated mean false alarm rates of 2.47% (SD = 8.88) in the absence of distractors and 6.36% (SD = 7.26) in the presence of distractors. False alarm rates significantly increased when multiple distractors preceded the cue (factor 'distractor number': $F(3, 78) = 3.10$, $p < .031$, $n_p^2 = .107$). Notably, the false alarm rate was on a comparably low level as in Kern and Niedeggen's [22] study (without distractors: M = 2.47%, SD = 8.13; multiple distractors: M = 3.48%, SD = 9.27).

The behavioral data from our experiment was compared to a previous cross-modal DIB study [22]: In the previous experiment, a simple local target feature change was used, whereas the current study includes a more complex dynamic display, and the target is defined by a more global change (see methods). Fig 4 illustrates the distractor effect in the complex as well as the simple target feature condition at SOA 0 ms.

The analysis unveiled a significant main effect of the 'target complexity' ($F(1, 61) = 4.33$, $p < .042$, $n_p^2 = .066$) with generally lower hit rates observed across distractor conditions in our

experiment (M = 85.83%, 95% CI [81.56, 90.10]), compared to the previous study
(M = 91.70%, 95% CI [88.00, 95.40]). As expected, there was a significant main effect of 'distractor number' (F(1.45, 88.37) = 52.40, p = .001, $n_p^2$ = .462). More importantly, the interaction between the within-subject factor 'distractor number' and the between-subject factor 'target complexity' at cue-target SOA 0 ms reached statistical significance (F(1.45, 88.37) = 3.79, p = .040, $n_p^2$ = .058). Additionally, the linear trend in the decreasing detection rate associated with an increasing distractor number seems to more expressed in the current dataset as compared to the previous study (F(1, 61) = 5.08, p = .028, $n_p^2$ = .077). While in both studies the distractor effect followed a significant linear trend (our data: F(1,26) = 31.42, p < .001, np2 = .547; 22: F(1,35) = 32.63, p < .001, np2 = .482), the drop in detection rate was more pronounced in the complex stimulus arrangement, with an averaged hit rate falling from 94% (0 distractors) to 74% (n distractors), compared to a less pronounced decrease from 96% (0 distractors) to 85% (n distractors) in the simple stimulus configuration [22].

## 3.2. Electrophysiological results

### 3.2.1. Distractor-evoked ERP signatures.
Grand-averaged ERP signatures of the frontal negativity are depicted in Fig 2C, corresponding mean amplitudes are presented in Fig 2D.

The distractor-evoked potential is defined by a focal negativity at about 90 ms, which is expressed lowest in the single distractor condition (M = -.23, SD = 1.09), gradually increasing during the presentation of multiple distractors (3–4 distractors: M = -.34, SD = 0.92; 5–6 distractors: M = -.79, SD = 1.02) (see Fig 2D). In line with this, the visual examination of the grand averages suggests a heightened negativity with a rising number of distractors (see Fig 2C). Furthermore, the negativity shares the spatial distribution of the distractor-related FN previously described [17, 20, 21] (see Fig 2B). Although no statistical main effect of the factor 'distractor number' was confirmed (F(2, 52) = 2.56, p = .087, $n_p^2$ = .089), the trend analysis still signaled a significant linear trend (F(1, 26) = 4.40, p = .046, $n_p^2$ = .145). The latter indicates a gradual linear progression from the FN's minimum expression in the single distractor condition to its most prominent expression in the highest distractor condition. A quadratic trend cannot be assumed (F(1, 26) = .60, p = .444, $n_p^2$ = .023). The frontal electrode cluster was homogenous with respect to this trend ('electrode' x 'distractor number': F(4.12, 106.98) = .792, p = .536, $n_p^2$ = .030).

### 3.2.2. Cue-target-evoked ERP signatures.
Grand averaged cue-target-evoked ERPs (P3, fronto-central negativity, posterior negativity) are depicted in Fig 3C and 3D.

The cue-target ERPs are characterized by an early fronto-central negativity (FCN) starting at 140 ms, reaching its peak at 240 ms, followed by a focal posterior negativity (PN) starting at 280 ms, peaking at 320 ms. The third component (labeled as P3) is a parietal positivity starting at 360 ms and lasting for 150 ms. Table 3 displays mean amplitudes of these components–for the P3, as well as the post-hoc identified fronto-central negativity, and the temporo-occipital negativity–across all three clusters.

The ERP data were analyzed in two steps: To identify whether a component was related to target processing, ERPs related to the conditions 'target present' (cue-target compound) and 'target-absent' (cue-only condition) were compared. In a second step, ERP components to a high and a low probability of target access were identified. To this end, the cue-target compound was segregated, and ERPs were averaged separately for the conditions with a "high" or "low" number of distractors.

Our a priori hypothesis was related to the P3 component: Against our hypothesis, visual inspection of grand-averaged potentials (see Fig 3C) implied no increase in amplitude, if the target was presented (cue-target: M = 4.43, SD = 3.46 vs. cue alone: M = 4.44, SD = 3.40).

**Table 3. Mean ERP amplitudes (in μV) and 95% confidence intervals (CIs) for cue-target-evoked electrophysiological components, separated for target absent vs. present.**

| | | ERP component | | |
|---|---|---|---|---|
| Condition | Electrode cluster | 140–200 ms (FCN) | 300–380 ms (PN) | 400–460 ms (P3) |
| target absent | fronto-central | **M = -.453** **[-.90, -.00]** | M = 1.565 [.43, 2.70] | M = 3.823 [2.47, 5.18] |
| | parietal | M = -.504 [-.94, -.07] | M = 1.577 [.61, 2.54] | M = 4.443 [3.10, 5.79] |
| | posterior | M = -.458 [-.83, -.09] | **M = .278** **[-.42, .98]** | M = 1.920 [.97, 2.87] |
| target present | fronto-central | **M = -1.207** **[-1.74, -.68]** | M = 1.961 [.84, 3.09] | M = 4.629 [3.25, 6.01] |
| | parietal | M = -.939 [-1.41, -.46] | M = .839 [-.25, 1.93] | M = 4.431 [3.07, 5.80] |
| | posterior | M = -.276 [-.74, .19] | **M = -1.72** **[-2.51, -.92]** | M = .798 [-.05, 1.65] |

Descriptive values for the exploratory fronto-central negativity (FCN) between 90 and 170 ms and posterior negativity (PN) between 300 and 380 ms, as well as the a priori predicted P3 between 400 and 460 ms, depicted separately for the fronto-central, parietal, and posterior cluster, as well as the target absent vs. target present condition. A significant effect of the factor 'target presence' was found for the FCN in the fronto-central cluster as well as for the PN in the posterior cluster (printed in bold).

Pursuant to this, the statistical analysis revealed no significant difference between both experimental conditions, with no main effect of target presence ($F(1, 26) = .00$, $p = .983$, $n_p^2 = .000$). Although there appears to be a significant interaction between the electrode position within the parietal cluster and the effect of target presence ('electrode' x 'target presence': $F(2.53, 65.85) = 8.79$, $p < .001$, $n_p^2 = .253$), the post-hoc comparisons did not reveal a selective effect on single electrodes (see S3 Table).

To approach the question whether cue-target ERPs are related to the detection performance, we calculated the ERP trials separately for experimental conditions defined by a low distractor number (0 or 1)–and a correspondingly higher detection rate–as well as a high distractor number (3–6), and a significantly reduced detection rate. This approach signaled that the P3 amplitude is not enhanced in the condition with a low distractor number: In the parietal electrode cluster, amplitudes were at a comparable level (high number of distractors: $M = 4.95$, $SD = 3.80$; low number of distractors: $M = 3.91$, $SD = 3.77$; see Table 4), and not significantly different ($F(1, 26) = 3.16$, $p = .087$, $n_p^2 = .108$). Again, a significant interaction was found between the factors 'electrode' and 'distractor number' ($F(2.87, 74.56) = 3.54$, $p = .020$, $n_p^2 = .120$). The post-hoc test revealed significant differences for the adjacent electrodes Pz, P3, P4—with a higher amplitude in the multiple distractor condition, contrary to what would have been expected a priori—but no significant differences for the more-anterior electrode pair CP1 and CP2 (S4 Table). This indicates a heterogeneity of the cluster, and that the effect is selectively expressed at posterior leads.

Although our hypothesis was clearly focused on the P3 effect, the grand-averaged ERP indicated that target processing and access might be associated with other ERP components. Since these components have been observed in previous experiments [23, 24], we added them in an a posterori analysis.

A fronto-central negativity was also observed in previous DID experiments [23, 24]. The higher amplitude in the 'cue-target' condition compared to the 'cue only' condition indicated that the component is related to the target processing (cue-target: $M = -1.21$, $SD = 1.34$ vs. cue alone: $M = -.45$, $SD = 1.14$). In line with the visual inspection (see Fig 3C), statistical analysis of

**Table 4. Mean ERP amplitudes (in μV) and 95% confidence intervals (CIs) for cue-target-evoked electrophysiological components, separated for a low vs. high number of distractors and therefore a manipulated target detection probability.**

| Condition | Electrode cluster | ERP component | | |
| --- | --- | --- | --- | --- |
| | | 140–200 ms (FCN) | 300–380 ms (PN) | 400–460 ms (P3) |
| low distractor number | fronto-central | M = -1.453 [-2.17, -.74] | M = 1.289 [.18, 2.40] | M = 4.463 [2.96, 5.96] |
| | parietal | M = -1.283 [-1.99, -.57] | M = .059 [-1.05, 1.17] | M = 3.912 [2.43, 5.40] |
| | posterior | M = -.487 [-1.06, .08] | **M = -2.485 [-3.44, -1.53]** | M = .131 [-.83, 1.09] |
| high distractor number | fronto-central | M = -.960 [-1.59, -.33] | M = 2.633 [1.22, 4.04] | M = 4.794 [3.33, 6.26] |
| | parietal | M = -.594 [-1.17, -.02] | M = 1.619 [.30, 2.94] | M = 4.950 [3.45, 6.45] |
| | posterior | M = -.064 [-.64, -.51] | **M = -.952 [-1.72, -.18]** | M = 1.465 [.55, 2.38] |

Descriptive values for the exploratory fronto-central negativity (FCN) between 140 and 200 ms and posterior negativity (PN) between 300 and 380 ms, as well as the a priori predicted P3 between 400 and 460 ms, depicted separately for the fronto-central, parietal, and posterior cluster, as well as the low vs. high distractor number condition. A significant effect of the factor 'distractor number' was found for the PN in the posterior cluster (printed in bold).

the time window 140–200 ms within the fronto-central cluster revealed a significant main effect of target presence ($F_{(1, 26)} = 6.82$, $p = .015$, $n_p^2 = .208$). This effect was not modulated by electrode position ('electrode' x 'target presence': $F_{(2.34, 60.87)} = .90$, $p = .426$, $n_p^2 = .033$). If the fronto-central negativity was separately analyzed for the distractor number conditions, the negativity appears to be shifted if less distractors were presented (high number of distractors: M = -.96, SD = 1.60; low number of distractors: M = -1.45, SD = 1.81; see Table 4 or Fig 3D). However, the rmANOVA did not signal a significant difference ($F_{(1, 26)} = 1.45$, $p = .240$, $n_p^2 = .053$). Moreover, the effect was also not differently expressed at single electrodes within the cluster ('electrode' x 'distractor number': $F_{(2.16, 56.19)} = 1.00$, $p = .381$, $n_p^2 = .037$).

The grand averages are also characterized by an occipital negativity between 300–380 ms. Its amplitude is more expressed when the target was present (M = -1.72, SD = 2.01), to when it was absent (M = .28, SD = 1.77), and this impression was confirmed by the rmANOVA with a highly significant main effect of target presence ($F_{(1, 26)} = 21.90$, $p < .001$, $n_p^2 = .457$). The significant interaction with the factor 'electrode' ('electrode' x 'target presence': $F_{(2.04, 53.14)} = 4.67$, $p < .013$, $n_p^2 = .152$) signaled that the difference is more strongly expressed at single electrodes within the cluster. Our post-hoc analysis indicates that the enhancement in case of a present target is expressed strongest at the more posterior electrodes O1 and O2 (see S5 Table).

In contrast to the other components, the expression of the occipital negativity appears to be related to the detection performance: If the cue-target ERPs were separately analyzed for the distractor number conditions, amplitudes were found to be less expressed when multiple distractors were present (M = -.95, SD = 1.94) as compared to when none or only a single distractor was present (M = -2.49, SD = 2.41) (see Table 4 or Fig 3D). Statistical analysis corroborated this impression with a strongly significant main effect of number of distractors ($F_{(1, 26)} = 20.69$, $p < .001$, $n_p^2 = .443$). Furthermore, the interaction between the factors 'target presence' and 'electrode' reached statistical significance ($F_{(2.76, 71.81)} = 3.68$, $p = .019$, $n_p^2 = .124$). Post-hoc analysis revealed that the effect of distractor number was, again, more strongly expressed at the posterior electrodes O1 and O2, but additionally also at the more anterior, lateralized electrode P8 (see S6 Table).

## 4. Discussion

### 4.1. Summary

Corresponding to our first hypothesis, the behavioral results show that the DIB effect can be replicated in a cross-modal setting: In our experiment the target detection rate is significantly reduced by an increasing number of distractor episodes. As expected, this functional relationship is mostly expressed when cue and target are presented simultaneously. Furthermore, the data replicate that the cross-modal DIB effect extends to the visual feature 'shape'.

In line with our second hypothesis, a frontal negativity was identified for distractor-evoked potentials, and its amplitude increases linearly with an increasing number of distractors. In contrast to our third hypothesis, target processing and detection did not enhance P3 amplitude. Rather, an occipital negativity–not observed in previous unimodal studies–was related to the processing and access to the visual target. All results will be discussed in more detail in the following.

### 4.2. Behavioral data

Our data confirm the idea that target access critically relies on the number of distractor episodes presented beforehand. Moreover—as suggested in previous papers—the distractor-induced effect is not bound to a specific stimulus modality or stimulus feature [13, 15, 21–24], which seems congruent to the notion that the presentation of multiple distractors triggers a central gating process [17, 20]. Target stimuli get involuntarily suppressed or inadequately processed. A consequence which can also be observed in related phenomena, such as contingent attentional capture (CAC; [39, 40]), repetition blindness (RB; [41, 42]), or the attentional blink (AB; [43, 44]). However, our data is further indication that the DIB effect can be contrasted to these phenomena.

In CAC, a single distractor preceding the target hinders processing of the target [45]. By using audiovisual stimuli, Dalton and Spence [46] were able to show that this effect even extends cross-modally. Our data, however, are more in line with previous findings suggesting that, in DIB, a single distractor does not have significant impact on the processing of an upcoming target [28]: As shown in Fig 4, the effect on target detection is fully expressed, if multiple distractors are presented. More importantly, the CAC effect is primarily related to a delay in the response to the target, but not to a lack in accessibility.

RB refers to the inability to report the second occurrence of a word (or picture) presented twice in an RSVP stream [41, 42]. This discrepancy has been attributed to a lack of token individuation [47], or an 'offline' retrieval problem [48]. A crucial factor for RB to occur is the physical resemblance between the two critical stimuli. Likewise in DIB, target processing is impeded solely by the presence of similar distractors, rather than dissimilar or absent ones [15]. Despite this evident similarity between both phenomena, research indicates that, in RB, the mere repetition of the target item once is sufficient to induce an effect. As discussed above, such an experimental condition seems not adequate to elicit DIB [28]. This difference between the two phenomena may be attributed to the distinct choice of physical stimuli: In RB, highly complex semantic stimuli are employed, requiring token individuation. In contrast, DIB involves stimuli comprising basic visual and/or auditory features that do not necessitate semantic-level processing.

In AB, conscious access to the second of two targets is constrained based on the temporal proximity to a single preceding target in an RSVP stream [49]. In contrast to distractor-induced effects mostly expressed at a cue-target SOA of 0 ms—a characteristic evident in our data as well—AB exhibits its maximum impact at a target-target SOA around 300 ms [50, 51].

Moreover, the AB was not reported consistently in cross-modal settings [52], whereas distractor-induced effects seem to be reliably expressed (for cross-modal DIB, see: 22, for cross-modal DID, see: 24).

Despite the replicability of the cross-modal DIB effect (see: 22), the direct comparison of the datasets also revealed significant differences: The expression of the effect in our experiment was significantly enhanced (25%) when contrasted to a previous study (9–14%; [22]). Fig 4 offers an illustration of this, depicting a steeper linear decrease in detection rates for the more complex visual stimulation implemented by us. In the previous study [22], the rather small expression of the DIB was attributed to a higher efficiency of auditory cues in resolving the central inhibition process. The current data rather indicates that the distractor-induced process is more expressed if more-complex visual stimuli are used, enhancing the perceptual load: As noted above, visual stimuli were previously defined by the clockwise motion of a preloader signal. This contrasts the two superimposed motion planes used in the current study.

A similar conclusion has previously been drawn for the unimodal DIB effect [14]: A higher perceptual load, which was achieved through an increased difficulty of detecting a visual cue, also lead to a higher impact of the visual distractors. The authors attributed this to an overall strengthened distractor inhibition under higher perceptual load, serving a maximized selection efficiency. More recent research has honed in on the connection between inhibitory processes and perceptual load: The findings suggest that under high perceptual load, the allocation of resources likely entails a gain increase for target information, coupled with an enhanced inhibition of distracting information [53].

Consequently, the higher perceptual load in the recent experiment may have facilitated a more focused allocation of attentional resources, thereby promoting a more efficient inhibition of distractors. The greater the efficiency of the inhibition, the greater the cost of resolving it, leading to an overall increased distractor effect, which we were able to measure in our data.

## 4.3. Distractor-evoked ERP signatures

Congruent with previous research, we identified an early frontal negativity which revealed a linear relationship with the number of distractors presented. The similarities in topography and functional behavior are in line with the previously identified frontal negativity [17, 20]. Earlier studies found that the increase in activation cannot be triggered by deviant stimuli embedded in the pre-cue epoch [15, 16, 21], because, in contrast to distractors, deviant stimuli do not share the properties of the upcoming target. This rules out that systematic changes in the ERP are due to habituation processes.

The early onset of our FN candidate, at about 100 ms, is rather surprising. Please note that the onset in previous unimodal DIB studies has been reported between 225 and 325 ms [17, 20, 21, 26]. This triggers two questions: Firstly, does the early negativity identified in the current study reflect a different process–not related to the inhibition of distractors? And secondly, are there differences in the setup which might explain such a drastic shift in latency?

When examining other early-onset frontal ERP components, the mismatch negativity (MMN) class emerges as a plausible candidate. Reported for the first time in the auditory domain [29, 54], the visual counterpart (visual mismatch negativity, vMMN; [55]) reveals a comparable onset latency between 100 and 180 ms [56]. The vMMN is proposed to mirror a prediction error, occurring when incoming sensory input diverges from the brain's predictions. In this conceptual framework, anticipations are automatically formed through the identification of trends present in recent sensory stimulation [29, 57–59]. Although a comparable process might be triggered by the successive presentation of distractors, it is unlikely that the frontal negativity observed in the recent study corresponds to a vMMN: While our ERP

signature is primarily evident over frontal electrode leads, vMMN typically manifests itself across occipital sites [56].

The discrepancy in spatial distribution does not only apply to the class of mismatch negativities, but extends to the category of awareness negativities [60, 61], including the visual awareness negativity (VAN), which, despite an early onset at about 200 ms, exhibits a posterior distribution [62–66]. The associated frontal negativity, known as auditory awareness negativity (AAN), is exclusively linked to the detection and discrimination of relevant auditory stimuli [30, 31, 67].

If we consider that the frontal negativity reflects a specific response to distractors, and signals the activation of a negative attentional set, as reported in previous papers [17, 20], the early onset of the signal reflects the requirements of the cross-modal setting: In contrast to the previous unimodal DIB-studies, attentional resources in the visual domain can be focused on changes in the dynamic random dot pattern and are not to be 'shared' with the monitoring of a second visual stream including the transient cue. The distribution of attentional resources to two modalities might affect the processing speed in the visual domain: Haroush et al. [68] observed that auditory processing is actually enhanced during visual AB. The authors propose that multimodal attentional resources might be liberated rather than engaged during cross-modal AB. In the same line, Regenbogen et al. [69] reported a clear susceptibility of fundamental auditory processing to cross-modal (visual) working memory load manipulations. The authors assume that in a cross-modal setup, top-down control necessary to inhibit a secondary task is reduced. This opens up the possibility that in a complex cross-modal setting, distractors might have a more direct access to the negative attentional set formed by the brain. However, it is essential to emphasize that the more direct path of distractors to influence the inhibition process does not necessarily indicate an overall stronger cumulative inhibition.

### 4.4. Cue-target-evoked ERP signatures

The most consistent finding in previous unimodal DIB studies has been that the processing and accessing of the target were signaled by an enhanced P3 amplitude [18, 20]. In our present cross-modal DIB study, however, this signature was not found to be related to the target: The P3 amplitude was not enhanced in the cue-target compound, when contrasted to the cue-only ERP. Furthermore, only the more posterior electrodes of the parietal cluster were modulated by the probability of successful target access. This modulation was stronger expressed following the presentation of multiple distractors, a scenario which can be associated with a reduced likelihood of successful target access. Hence, an outcome contrary to our expectations. Inspection of Fig 3D suggests that the difference found at posterior parietal leads is primarily due to the preceding expression of the occipital negativity.

Our results parallel findings from a previous ERP experiment on cross-modal DID [24], and question the idea that the P3 reflects a critical cognitive process related to target access, such as updating in working memory [70, 71]. In fact, task-irrelevant stimuli do not elicit ERP signatures that belong to the class of P3 components, even if participants are clearly aware of them [72]. Furthermore, data from neurological patients revealed that the P3b response can be absent, even if patients are aware of a target stimulus [73, 74]. In sum, it is questionable to assume that the P3 serves as a sufficient indicator for visual awareness [61, 75]. Our results are rather in line with studies suggesting that the P3 is related to post-perceptual processes. A solid body of literature links the component to decision-making and target report [72, 76–78]. Consistent with this concept, studies have shown that the amplitude of the P3 increases with an increasing certainty of participant's target-related decisions [79, 80].

However, in our data, an occipital ERP component was found to be rather related to the processing of the target, as well as to the probability of correctly reporting the target. The

topography of the posterior negativity, its latency, and functional characteristics share the properties of the previously reported VAN [62], which has been observed for various visual stimulus manipulations [65, 66, 81]. As mentioned above, the class of awareness negativities (AAN: [30, 82], SAN: [83]) has been related to the (neural) emergence of stimulus awareness (for an opposing attentional account, see: [84]).

Notably, a posterior negativity serving as a marker for target access has not been reported in previous **unimodal** DIB studies [17, 20, 21, 26]. The lack of a modulation of early visual ERP components lead the authors to the conclusion that the DIB effect relies on a mere post-perceptual process [18]. The current cross-modal data, however, suggests that access to targets restricted by prior distractors may rely on early visual system activation (reflected by the VAN). The absence of this ERP component in earlier unimodal studies may be due to an over-shadowing effect from concurrent processing of the equally visual cue. Replacing the visual cue by an auditory one reveals the discernibility of the VAN in the context of target processing. This suggests expanding the assumption, that the cue merely serves as a temporal marker resolving attentional inhibition [14]: The congruency of the cue with the target's modality appears to be an important experimental factor. To emphasize this fact, we decided to label the DIB effect associated with an auditory cue as a 'cross-modal' setup. Furthermore, this labeling is consistent with the labeling used in previous studies [22, 24].

Our findings align with the general idea that, when exposed to multisensory stimulation, neural responses differ significantly compared to responses observed during unisensory stimulation [85–87]. What is more, in a study by Filimonov et al. [88], the authors suggest that VAN and AAN serve as modality-specific early indicators of visual and auditory awareness, whereas the P3 complex exhibits more modality-general characteristics. We therefore tentatively propose the additional idea that the cross-modal setup facilitates modality-specific processing of the visual stream, whereas the unimodal setup is predominantly associated with modality-general processes.

To delve deeper into the observed discrepancy between uni- and cross-modal findings, exploring neural correlates of target awareness in cross-modal experiments combining even more senses (e.g., pairing tactile cues with visual or auditory targets/distractors) could be a fruitful avenue for future research. If the pivotal element should indeed be that the cue differs in modality from the target/distractors, we anticipate early negativities as reliable indicators of awareness in these scenarios as well.

### 4.5. Limitations

The interpretation of the current dataset is limited by a number of factors.

First of all, target awareness was not directly related to ERPs, as done in earlier DIB studies (see: 20, 23, 24), but only indirectly (low vs. high number of distractors). This is due to the fact that the study design also incorporated the investigation of distractor processing. Although this leads to a more conservative interpretation of cue-target ERPs, it makes the comparison with previous results more complicated. Future studies might also consider that the functional role of the P3 can be rather identified by using a rating system regarding perceptual awareness decisions, thereby taking into account that consciousness can be conceptualize as a graded experience [89, 90].

Furthermore, our study is the first to use a fixed probe position with the intention to exclude temporal expectations. These have been a problem in earlier experiments (e.g. 20). Previously, the expectation of the cue increased in parallel with the length of the pre-cue epoch. Such an influence on potentials may result in 'late distractors' appearing more pronounced in the EEG signal when compared to 'early' or 'medium distractors', simply due to

their proximity to the cue. Our data rules this out, since participants' expectations are held constant. Eliminating this confounding variable benefits our experimental design but complicates comparisons with previous unimodal DIB research on ERP responses to different temporal probe positions [17, 20, 21].

What is more, we did not control for inter-individual differences in our participants. As shown for the AB, differences in executive working memory functioning might contribute to the performance during visual detection tasks [91]. Correspondingly, Milders and colleagues [19] reported that, during DIB, low Stroop interference scores correlate with a higher distractor effect. Kern and Niedeggen [23, 24] implied the existence of inter-individual differences also regarding uni- and cross-modal DID. Hence, it is conceivable that high-level cognitive abilities modulate distractor-induced blindness and may have exerted an influence on our findings, which needs to be followed up in future studies.

Another noteworthy limitation of our study, pertaining to the above mentioned factor, is the potential bias stemming from the selection of participants, as our sample exclusively comprises psychology students. This homogeneity may limit generalizability of our findings to a broader population since people with a higher academic background might possess unique mental processing skills [92–94]. However, the above described link between stronger distractor effects and higher executive functions implies that the DIB effect might be more pronounced in our sample than in the general public, facilitating a detailed examination of its characteristics.

Additionally, we would like to draw attention to the following issue: In the present study, the term 'cross-modal' specifically refers to the incongruent modality between target and cue, rather than target and distractors. Previous research has demonstrated that a crucial prerequisite for the DIB phenomenon to arise is the presence of distractors that share the visual features of the target (Michael et al., 2011; Michael et al., 2012). Given the significant differences in properties between auditory and visual stimuli, no blindness effect is expected to occur, if auditory distractors were paired with a visual target. Nevertheless, while in our setup blindness is triggered by visual distractors, the experimental task still requires simultaneous monitoring of two streams located in different modalities. Both streams contain stimuli necessary for successful task completion. With the cue and the target potentially appearing in parallel, it can be assumed that attention is divided between the two streams. What is more, our results indicate that monitoring the auditory stream in addition to the visual one seems to have an influence on the underlying neural processes detected in the EEG signal in response to the distractors as well as the target. However, whether these deviations, specifically regarding the distractor-evoked potentials, stem from a general effect of cross-modality, remains unclear. To elucidate this, it will be necessary to examine distractor-evoked potentials from the opposing cross-modal direction–in response to auditory distractors paired with an auditory target and a visual cue. While such results are still forthcoming, we preliminary advocate for using the label 'cross-modal' in the present study. Taken together, we interpret our results, along with previous findings, as evidence for a significantly different processing when cue and target are defined in incongruent modalities.

An alternative approach to multimodal research regarding the DIB/DID paradigm could involve equipping target and distractors with auditory as well as visual features at the same time. If the distractors share the target's visual as well as auditory properties, a blindness/deafness effect is expected to occur. This method would emphasize the modality of interest between distractors and target in a more integrative manner, focusing on how shared features across modalities influence the phenomenon. However, research by Michael and colleagues (2012) from the visual domain suggests that, even in a combined-feature task, feature representations are expected to remain separate.

Lastly, it is important to emphasize that, with respect to the cue-target related ERPs, the most-sensitive component in our analysis (VAN) was identified a posteriori, predominantly driven by our data. This should, hence, be viewed with caution and warrants a replication of the effect.

## 5. Conclusion

The current study demonstrates that the distractor-induced effects can be observed in a cross-modal setting. Furthermore, a gradual frontal activation is triggered with an increasing number of visual distractors which supports the idea of a central inhibition process. However, the impact of a divergent modality of the cue is likely signaled by a more direct path of distractors towards the inhibition, as well as the aberrant ERP responses to the cue-target complex: The cross-modal setup might uncover the impact of sensory cortices in target awareness—not observed in the unimodal scenario, yet.

In sum, these results substantiate the assumption of an attentional system which is fundamentally influenced by multimodal processing demands [95]. As proposed by Talsma and colleagues [96], the complexity of the environmental stimulation might play a crucial role in determining how attention and multisensory processing interact. To tackle this in more detail, a systematic analysis comparing our findings with distractor-evoked potentials observed during cross-modal DID is required. Such an investigation would aid in determining whether similar attentional mechanisms operate across both cross-modal directions. Overall, cross-modal processing clearly transcends a simple electrophysiological reflection of its unimodal counterpart. For future studies aiming to contribute insights into human perception in complex, multimodal environments, considering this is essential.

## Supporting information

**S1 Fig. Graphical representation of the mean amplitudes for the distractor-evoked ERP following both liberal and conservative artifact rejection, categorized by distractor condition.**
(PDF)

**S1 Table. Mean target detection rates for the pilot as well as for the present study, based on 20 trials per experimental condition.**
(PDF)

**S2 Table. Mean amplitudes for the distractor-evoked ERP after the liberal and conservative artefact rejection per distractor condition, including the corresponding number of trials they were based on.**
(PDF)

**S3 Table. Post hoc paired t-tests for the five electrodes of the parietal cluster, comparing the conditions 'cue-target' vs. 'cue-only'.**
(PDF)

**S4 Table. Post hoc paired t-tests for the five electrodes of the parietal cluster, comparing the conditions 'low' vs. 'high' distractor number.**
(PDF)

**S5 Table. Post hoc paired t-tests for the five electrodes of the posterior cluster, comparing the conditions 'cue-target' vs. 'cue-only'.**
(PDF)

**S6 Table. Post hoc paired t-tests for the five electrodes of the posterior cluster, comparing the conditions 'low' vs. 'high' distractor number.**
(PDF)

## Acknowledgments

We thank Jacob Wolfram, Jennifer Klinger, Sara Ilter, and Dominik Fall for their assistance in data collection. We furthermore thank Wolf Culemann for his expertise in coding and his help concerning stimuli generation.

## Author Contributions

**Conceptualization:** Sophie Hanke, Michael Niedeggen.

**Data curation:** Sophie Hanke.

**Formal analysis:** Sophie Hanke.

**Investigation:** Sophie Hanke.

**Methodology:** Sophie Hanke, Michael Niedeggen.

**Project administration:** Sophie Hanke, Michael Niedeggen.

**Resources:** Michael Niedeggen.

**Software:** Michael Niedeggen.

**Supervision:** Michael Niedeggen.

**Validation:** Michael Niedeggen.

**Visualization:** Sophie Hanke.

**Writing – original draft:** Sophie Hanke.

**Writing – review & editing:** Sophie Hanke, Michael Niedeggen.

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
