## [Decision Letter · Decision Letter 0]

16 May 2024

PONE-D-24-10062Event-related potentials of stimuli inhibition and access in cross-modal distractor-induced blindnessPLOS ONE

Dear Dr. Hanke,

Thank you for submitting your manuscript to PLOS ONE. After careful consideration, we feel that it has merit but does not fully meet PLOS ONE’s publication criteria as it currently stands. Therefore, we invite you to submit a revised version of the manuscript that addresses the points raised during the review process.

Your manuscript has been reviewed by three experts in the field. As you will see below, all three reviewers agree on the technical quality of your contribution but raise some conceptual points which you might find helpful. Specifically, I agree with Reviewer #1 that the relationship between DIB and similar effects such as repetition blindness could be discussed in more detail. I also share with Reviewers #2 and #3 the concern that your paradigm is largely unimodal in the sense that the distractors and targets were both presented in the visual modality and only the cue was crossmodal. I think that addressing these points could strengthen the theoretical motivation of your manuscript, and therefore encourage you to revise your manuscript along the lines suggested by the reviewers. I expect that a final decision on this manuscript can likely be reached after this round of revisions.

We look forward to receiving your revised manuscript.

Kind regards,

Patrick Bruns

Academic Editor

PLOS ONE

Journal Requirements:

2. Please remove your figures from within your manuscript file, leaving only the individual TIFF/EPS image files, uploaded separately. These will be automatically included in the reviewers’ PDF.

Reviewers' comments:

Reviewer's Responses to Questions

**Comments to the Author**

1. Is the manuscript technically sound, and do the data support the conclusions?

Reviewer #1: Yes

Reviewer #2: Partly

Reviewer #3: Yes

2. Has the statistical analysis been performed appropriately and rigorously? 

Reviewer #1: Yes

Reviewer #2: Yes

Reviewer #3: Yes

3. Have the authors made all data underlying the findings in their manuscript fully available?

Reviewer #1: Yes

Reviewer #2: Yes

Reviewer #3: Yes

4. Is the manuscript presented in an intelligible fashion and written in standard English?

Reviewer #1: Yes

Reviewer #2: Yes

Reviewer #3: Yes

5. Review Comments to the Author

Reviewer #1: Summary: The authors describe an experiment in which they investigate ERP correlates of a phenomenon called „distractor-induced blindness“ (DIB) in a multi-modal task. The term DIB reflects impaired detection of a target when it is preceded by a series of similar distractors relative to a condition in which the target is not preceded by distractors. In the typical DIB task, the target is signaled by a cue, which means that participants have to detect both the cue and the subsequent target. In the experiment described here, the distractor and target events consisted in brief changes of the shape of small visual elements that were moving across a computer screen. The cue was a short increase in a continuous tone presented to the left ear, while irrelevant tones were also presented to the right ear. The independent variables were the number of distractors that preceded a target (0, 1, 3-4, 5-6), and cue-target SOA (0 or 300 ms). Behavioral results replicated the DIB effect in a multimodal setting: An increasing number of distractors decreased target detection rate. With regard to the ERP correlates of distractor processing, the results also replicated previous findings: The amplitude of a frontal negativity increased linearly with an increasing number of distractors. With regard to the ERP correlates of cue-target processing, the results were somewhat unexpected. In contrast to previous findings from unimodal tasks, target processing and detection were not reflected in P3 amplitude, but in an occipital negativity.

Evaluation: This paper describes a competently designed experiment on an interesting and timely issue, which produced some interesting findings. Therefore, I do not have fundamental objections against publication of this manuscript. Yet, I think there is some room for improvement with regard to the description of the DIB, and with regard to discussing the relationship between the DIB and similar phenomena such as repetition blindness.

Specific comment: The so-called DIB effect basically consists in the observation that the processing of a target is impaired when it is preceded by one or more similar distractors relative to a condition in which the target is not preceded by similar distractors. Hence, if I understood correctly, target-distractor similarity seems an important ingredient to the DIB effect. In fact, the literature contains several types of similar effects in which the processing of targets is impaired by the presentation of similar distractors relative to dissimilar or no distractors. Among these phenomena are (a) negative priming, (b) repeated-letter inferiority (e.g., Bjork & Murray, 1977), and (c) repetition blindness (e.g., Arnell & Shapiro, 2011; Kanwisher, 1987). The authors are discussing relationships between DIB and attentional blink, and between DIB and contingent attentional capture, but I think that repetition blindness is more relevant here. The reason is that target-distractor similarity is neither a crucial factor for attentional blink nor for contingent-attentional capture, but target-distractor similarity is a crucial factor in repetition blindness. Therefore, I suggest to explain the role of target-distractor similarity for the DIB in more detail, and to discuss the relationship (i.e., similarities and differences) between DIB and repetition blindness.

Stylistic suggestions:

• Page 15, lines 362-363: The sentence „Correspondingly to the definition (…“ seems incomplete.

• Page 24, line 565: „stating“ should be „starting“.

References

Arnell, K. M., & Shapiro, K. L. (2011). Attentional blink and repetition blindness. WIREs Cognitive Science, 2(3), 336–344.

Bjork, E. L., & Murray, J. T. (1977). On the nature of input channels in visual processing. Psychological Review, 84(5), 472–484.

Kanwisher, N. G. (1987). Repetition blindness: Type recognition without token individuation. Cognition, 27(2), 117–143.

Reviewer #2: The research used EEG to study the distractor-induced blindness effect. The DIB/DID previously has been studied in both uni-modal and cross-modal situations, with the cross-modal study using auditory stimuli. The present study extended the cross-modal research by switching to visual stimuli. Different from the predictions based on the previous cross-modal DID study, the current findings did not find several predicted ERP components.

As a disclaimer, I am not familiar with the DIB effect before reading this article. Also, the several ERP components the present study analyzed are not usually concerned in my EEG research. Given these, I’d only raise a few comments regarding the big picture of the study, that are important for the research to make sense to broader audience.

Based on my understanding of the approach, what were manipulated “cross-modal” in the DIB paradigm was actually the cue vs. the target and distractors. While what DIB truly concerns is the effect of target-like distractor exposure (IV) on target identification (DV), I feel that the cue, as a non-critical variable, plays a relatively small role in this effect. Maybe the authors could elaborate on why it is necessary to vary the modality of the cue, and any implications from the cross-modal study above and beyond the uni-modal study of this effect. This would establish the foundation for using EEG to study such an effect in the first place.

On Page 11 and 14, the number of trials in each cell condition seems too small to me, both in terms of obtaining a reliable mean behavioral target detection response and reliable averaged ERP waveforms. Although the behavioral results even showed an increase in the DIB effect compared to previous studies, the small number of trials in ERP averaging made it questionable that the several null effects in ERP analysis could be due to relatively large variability without an ample number of trials. The lack of a replication experiment further exacerbates my concern on the interpretation of the null effects. The number of trials used by earlier experiments that successfully detected those ERPs of interest may be cited in support of the current design.

Other minor issues:

1. Some concepts mentioned in results and discussion were not corresponding introduced in the Method section, such as “target complexity” and “perceptual load”. It is possible that the authors framed the terms differently in wording the manipulations, which seems lack of consistency for understanding the article.

2. When reporting the ANOVA results, the degrees of freedom in F-statistics were decimals in several places. This may indicate unequal variances in the data. The authors may want to double check if any assumptions of the statistical test were violated.

Reviewer #3: I have to confess I am not very deep into the DIB literature but I am working on the field of distractor processing in general. So I can’t not precisely judge the theoretical gain of this study – to me it seems mediocre at theoretical grounds. But as I understand the politics at PlosOne that is fine as long as the study is lege artis concerning the methods & results and it is. The data replicate previous findings except for the occipital activation (the third hypothesis). Yet, the authors give some good arguments/interpretations why this is the case.

I found the fact that the authors see this as an instance of cross-modal DIB somewhat disturbing because in my mind the modality of the cue is not so relevant. So when I first read about cross-modal DIB I was of course thinking about distractors and targets being presented to different modalities – yet, this is only personal taste, I guess.

6. PLOS authors have the option to publish the peer review history of their article (what does this mean?). If published, this will include your full peer review and any attached files.

Reviewer #1: No

Reviewer #2: No

Reviewer #3: No

---

## [Author Response · Author response to Decision Letter 0]

19 Jun 2024

Dear Dr. Patrick Bruns,

Please find enclosed our revised manuscript (Event-related potentials of stimuli inhibition and access in cross-modal distractor-induced blindness) submitted for your consideration for publication in PLOS ONE. 

First and foremost, we would like to extend our gratitude to you and the reviewers for your insightful comments. We have carefully considered all feedback, which has significantly enhanced the quality and comprehensibility of the manuscript. 

In response to the reviewers' points, we have thoroughly addressed each one: Specifically, we have elaborated on the relationship between distractor-induced blindness (DIB) and the conceptually related effect of repetition blindness. Furthermore, we trust that our detailed explanations and the amendments made to the manuscript will convince you and the reviewers that our paradigm can indeed be accurately described as 'cross-modal'.

The corresponding changes in the revised manuscript (see: Revised Manuscript with Track Changes) are detailed below, point by point. We hope that these substantial revisions and additions will meet the reviewers' expectations and that the improved quality of the manuscript will be sufficient for publication in your esteemed journal.

Sincerely,

Sophie Hanke (on behalf of the authors)

Reviewer #1: 

Thank you very much for your detailed and valuable feedback. Below you will find our responses to your comments.

The authors are discussing relationships between DIB and attentional blink, and between DIB and contingent attentional capture, but I think that repetition blindness is more relevant here. The reason is that target-distractor similarity is neither a crucial factor for attentional blink nor for contingent-attentional capture, but target-distractor similarity is a crucial factor in repetition blindness. Therefore, I suggest to explain the role of target-distractor similarity for the DIB in more detail, and to discuss the relationship (i.e., similarities and differences) between DIB and repetition blindness.

Thank you for your comment. We have thoroughly reviewed the phenomenon of repetition blindness (RB; Arnell & Shapiro, 2011; Coltheart, 2010), which refers to the inability to report the second of two targets (either words or pictures) presented in a RSVP stream, if the same target has been presented previously.

After careful consideration, we came to the conclusion that the same difference between DIB and contingent attentional capture, already highlighted in the original manuscript, also accounts for RB: Despite the obvious similarity to the DIB effect concerning the physical resemblance between distractor and target stimuli, previous DIB results revealed that a single distractor is not sufficient to elicit the effect, but that the effect requires the presentation of multiple distractors (Winther & Niedeggen, 2017b). In RB, the simple repetition of the target item is sufficient to elicit the effect. This difference might (also) be attributed to the choice of the physical stimuli: In RB, highly complex semantic stimuli are used, and a token individuation is assumed to be required (Kanwisher, 1987). In DIB, stimuli are very simple visual and/or auditory features which are not to be processed on a semantic level.

We refer to the difference in the paradigms in the revised manuscript:

Page 30: “RB refers to the inability to report the second occurrence of a word (or picture) presented twice in an RSVP stream (Arnell & Shapiro, 2011; Coltheart, 2010). This discrepancy has been attributed to a lack of token individuation (Kanwisher, 1987), or an ‘offline’ retrieval problem (Whittlesea & Masson, 2005). A crucial factor for RB to occur is the physical resemblance between the two critical stimuli. Likewise in DIB, target processing is impeded solely by the presence of similar distractors, rather than dissimilar or absent ones (Michael et al., 2011). Despite this evident similarity between both phenomena, research indicates that in RB the mere repetition of the target item once is sufficient to induce an effect. As discussed above, such an experimental condition seems not adequate to elicit DIB (Winther & Niedeggen, 2017b). This difference between the two phenomena may be attributed to the distinct choice of physical stimuli: In RB, highly complex semantic stimuli are employed, requiring token individuation. In contrast, DIB involves stimuli comprising basic visual and/or auditory features that do not necessitate semantic-level processing.” 

Stylistic suggestions:

• Page 15, lines 362-363: The sentence „Correspondingly to the definition (…“ seems incomplete. 

• Page 24, line 565: „stating“ should be „starting“.

Thank you kindly for highlighting these stylistic aspects that we missed during our initial proofreading. Both amendments have been incorporated into the revised manuscript.

Reviewer #2:

Thank you very much for your kind and thoughtful feedback. Please find our detailed responses, to all the points you raised, below.

Based on my understanding of the approach, what were manipulated “cross-modal” in the DIB paradigm was actually the cue vs. the target and distractors. While what DIB truly concerns is the effect of target-like distractor exposure (IV) on target identification (DV), I feel that the cue, as a non-critical variable, plays a relatively small role in this effect. Maybe the authors could elaborate on why it is necessary to vary the modality of the cue, and any implications from the cross-modal study above and beyond the uni-modal study of this effect. This would establish the foundation for using EEG to study such an effect in the first place.

Thank you for your comment. We have to admit that the motivation for the cross-modal approach was not sufficiently elucidated in our original manuscript. 

Congruent with the reviewer’s note, previous experiments on the visual DIB indicated that the cue merely serves as a release signal. The effect does not critically depend on the spatial position of the cue (peripheral (Niedeggen et al., 2004) vs. central stream (Milders et al., 2004; Niedeggen et al., 2002; Sahraie et al., 2001)). Moreover, the feature of the cue type does not modulate the effect (letter (Hay et al., 2006) vs. luminance change (Winther & Niedeggen, 2017a, 2018) vs. onset of ‘transparent’ motion (Niedeggen et al., 2004), vs. color change (Milders et al., 2004; Niedeggen et al., 2015; Niedeggen et al., 2012; Niedeggen et al., 2002; Sahraie et al., 2001; Winther & Niedeggen, 2017b). This point has been addressed in previous studies (Hay et al., 2006). Notably, the design remains unimodal with respect to cue and target in all studies. 

However, electrophysiological findings already suggested that the processing of the cue-target complex is different when cue and target were defined in distinct modalities (Kern & Niedeggen, 2021b): For unimodal DID (cue and target presented in the auditory domain), ERP responses to the cue-target compound were still characterized by a prominent P3 – if the target is detected successfully (Kern & Niedeggen, 2021a). Yet, correctly reported targets were additionally characterized by a stronger fronto-central negativity at about 200 ms. What is more, for the cross-modal DID effect (visual cue and auditory target), successful target detection was exclusively characterized by a pronounced fronto-central negativity, whereas the P3 amplitude was not enhanced when contrasted to misses (Kern & Niedeggen, 2021b).

These results suggests that the processing of the target is affected by the modality of the cue. From our point of view, this finding qualifies the task used in our study as a ‘cross-modal’ task. 

We have clarified this point in our revised paper:

Page 5: “In sum, these deviant ERP results signal that the central suppression process previously identified in uni-modal visual distractor studies must be questioned if cue and target are defined in different modalities. The processing of a target stimulus in a multiple-distractor task seems to be affected by the congruency of the cue’s modality. In this respect, we refer to studies in which cue and target are defined in the same modality as ‘unimodal’. The current study defines the cue and target in different sensory domains and is labeled ‘cross-modal’.”

Please note that the ERP results of our study also emphasize the role of the different cue/target modalities: The posterior negativity – not observed in the uni-modal visual DIB studies (Niedeggen et al., 2015; Niedeggen et al., 2004; Niedeggen et al., 2012; Winther & Niedeggen, 2017a) – has probably been masked by the choice of the visual cue. Finally, the label ‘cross-modal’ is consistent with the labeling used in the previous studies – based on a similar setup (Kern & Niedeggen, 2021b, 2023). 

Page 35: “Replacing the visual cue by an auditory one reveals the discernibility of the VAN in the context of target processing. This suggests expanding the assumption that the cue merely serves as a temporal marker resolving attentional inhibition (Hay et al., 2006): The congruency of the cue with the target's modality appears to be an important experimental factor. To emphasize this fact, we decided to label the DIB effect associated with an auditory cue as a ‘cross-modal’ setup. Furthermore, this labeling is consistent with the labeling used in previous studies (Kern & Niedeggen, 2021b, 2023).”

On Page 11 and 14, the number of trials in each cell condition seems too small to me, both in terms of obtaining a reliable mean behavioral target detection response and reliable averaged ERP waveforms. Although the behavioral results even showed an increase in the DIB effect compared to previous studies, the small number of trials in ERP averaging made it questionable that the several null effects in ERP analysis could be due to relatively large variability without an ample number of trials. The lack of a replication experiment further exacerbates my concern on the interpretation of the null effects. The number of trials used by earlier experiments that successfully detected those ERPs of interest may be cited in support of the current design.

We appreciate you bringing this important point to our attention and recognize that we did not communicate this information clearly enough in our original manuscript.

As for the ERP analysis, the number of trials available for the analysis of the distractor effect was sufficient: Please note that for each distractor condition (distractor number 1 vs. 3-4 vs. 5-6) a total of 55 trials per condition was foreseen. The ERP analysis of distractor effects in previous studies was based on a comparable number (n=63 trials per distractor condition; Niedeggen et al., 2012; Winther & Niedeggen, 2017a). As in previous experiments, this number also considered a rigorous artefact rejection, leading to the loss of approximately 50% of the trials (Kern & Niedeggen, 2021b: In the distractor-sensitive subgroup, a mean of 50.76 hit trials (SD=12.19) out of 80 trials remained after artifact rejection; In the distractor-insensitive subgroup, a mean of 38.77 hit trials (SD=9.97) out of 80 trials remained after artifact rejection). The rejection rate in the present study was comparable to that of previous studies, leading to an analogous signal-to-noise ratio of the averaged ERPs in single participants. 

This point has been clarified in the revised manuscript:

Page 11: “For the analysis of the distractor-evoked potentials, 55 trials were available for each distractor condition (1 vs. 3-4 vs. 5-6), with the SOA and control conditions combined. In previous studies, the analysis of distractor-evoked ERPs was based on a comparable number (n=63 trials per distractor condition; Niedeggen et al., 2012; Winther & Niedeggen, 2017a).”

As well as page 14: “The rejection rate in the present study was comparable to that of a previous experiment (Kern & Niedeggen, 2021b: In the distractor-sensitive subgroup, a mean of 50.76 hit trials (SD = 12.19) out of 80 trials remained after artifact rejection; In the distractor-insensitive subgroup, a mean of 38.77 hit trials (SD = 9.97) out of 80 trials remained after artifact rejection), leading to an analogous signal-to-noise ratio of the averaged ERPs in single participants.”

As for the behavioral analysis, the computation of the mean hit rate in each participant was based on 20 trials per distractor condition. This trial number was based in a pilot study, and we can take these results to estimate the reliability of the effect. 

S1 Table

Mean target detection rates for the pilot as well as for the present study, based on 20 trials per experimental condition.

Distractors Pilot study (n=15)

(SOA 0 ms condition) Present study (n=27)

(SOA 0 ms condition)

0 M = 90.45 M = 94.74

 CI [83.81, 97.08] CI [91.05, 98.44]

1 M = 87.16 M = 88.36

 CI [79.48, 94.84] CI [83.41, 93.31]

3-4 M = 75.81 M = 82.37

 CI [65.41, 86.20] CI [75.64, 89,10]

5-6 M = 70.25 M = 74,37

 CI [56.63, 83.86] CI [66.15, 82.59]

Note. Mean (in %) and the corresponding 95% confidence interval (CI) for averaged correct target detection after correct cue detection, reported separately for each distractor condition. SOA = stimulus onset asynchrony.

Both experiments, the pilot study as well as the present ERP study, clearly indicate that the hit rate is affected significantly by the number of distractors. For the pilot study, the effect of the distractor number was strongly expressed (F(1.76, 24.58) = 10.90, p <.001, np2 = .438) and followed a linear trend (F(1,14) = 17.48, p < .001, np2 = .555). Also for the present study (see manuscript page 22), the effect of the distractor number was strongly expressed (F(2.03, 52.89) = 20.92, p <.001, np2 = .446) and followed a linear trend (F(1,26) = 32.27, p < .001, np2 = .554). 

What is more, a combined analysis (mixed ANOVA) of both experiments highlights the reliability of the observed behavioral effect: No significant interaction between the within-subjects factor ‘distractor number’ (0 vs. 1 vs. 3-4 vs. 5-6) and the between-subjects factor ‘experiment’ (pilot vs. present experiment) could be found (F(3,120) = .44, p < .726, np2 = .011). 

In the revised manuscript, we now included that the estimation of trial number necessary to reveal a distractor effect was based on a pilot study, and that the results of the pilot study can be found in the supplement:

Page 11: “This number of trials was based on the outcome of a pilot study (n=15 participants), see supplement (S1 Table). Most importantly, the pilot data confirm the reliability of the effect of distractor number.”

Last but not least, we fully acknowledge the importance of replicating our results. Additionally, we have made every effort to avoid misinterpreting null results throughout the manuscript.

Some concepts mentioned in results and discussion were not correspondingly introduced in the Methods section, such as “target complexity” and “perceptual load”. It is possible that the authors framed the terms differently in wording the manipulations, which seems lack of consistency for understanding the article.

Thank you kindly for highlighting these inconsistencies about the wording used in our manuscript. Both terms were used to describe the visual setup defined by a transparent motion which is clearly different from the setup used in the preceding cross-modal study. To enhance clarity and consistency, we restricted ourselves to the term ‘complexity’, and added the following sentences to the Methods section:

Page 13: “In the current study, a more complex dynamic pattern was used (two superimposed motion planes). This setup has been previously established by Valdes-Sosa and colleagues (Pinilla et al., 2001; Rodríguez & Valdés-Sosa, 2006; Valdes-Sosa et al., 1998) and requires a visual segregation process. We will refer to this difference in visual stimuli configuration as ‘complexity’ in the following.” 

When reporting the ANOVA results, the degrees of freedom in F-statistics were decimals in several places. This may indicate unequal variances in the data. The authors may want to double check if any assumptions of the statistical test were violated.

Thank you for 

---

## [Decision Letter · Decision Letter 1]

8 Jul 2024

PONE-D-24-10062R1Event-related potentials of stimuli inhibition and access in cross-modal distractor-induced blindnessPLOS ONE

Dear Dr. Hanke,

Thank you for submitting your manuscript to PLOS ONE. After careful consideration, we feel that it has merit but does not fully meet PLOS ONE’s publication criteria as it currently stands. Therefore, we invite you to submit a revised version of the manuscript that addresses the points raised during the review process.

As you can see from the reviews below, Reviewer #2 still has concerns about the role of cue modality in your paradigm as well as about the reliability of the ERP waveforms. Please try your best to fully address these concerns with your next revision.

We look forward to receiving your revised manuscript.

Kind regards,

Patrick Bruns

Academic Editor

PLOS ONE

Journal Requirements:

Reviewers' comments:

Reviewer's Responses to Questions

**Comments to the Author**

1. If the authors have adequately addressed your comments raised in a previous round of review and you feel that this manuscript is now acceptable for publication, you may indicate that here to bypass the “Comments to the Author” section, enter your conflict of interest statement in the “Confidential to Editor” section, and submit your "Accept" recommendation.

Reviewer #1: All comments have been addressed

Reviewer #2: All comments have been addressed

2. Is the manuscript technically sound, and do the data support the conclusions?

Reviewer #1: Yes

Reviewer #2: Partly

3. Has the statistical analysis been performed appropriately and rigorously? 

Reviewer #1: Yes

Reviewer #2: Yes

4. Have the authors made all data underlying the findings in their manuscript fully available?

Reviewer #1: Yes

Reviewer #2: Yes

5. Is the manuscript presented in an intelligible fashion and written in standard English?

Reviewer #1: Yes

Reviewer #2: Yes

6. Review Comments to the Author

Reviewer #1: Summary and evaluation: This paper contains the revised version of a manuscript I have reviewed before. In their paper, the authors describe an experiment by which they investigate ERP correlates of a phenomenon called „distractor-induced blindness“ (DIB) in a multi-modal task. I already liked the previous version of the manuscript, although I saw some room for improvement with regard to the description of DIB, and with regard to discussing the relationship between DIB and similar phenomena such as repetition blindness (RB). The authors have added a short discussion of similarities and differences between DIB and RB to the GD of the revised manuscript, and they have corrected some stylistic issues. I am satisfied with the authors response to my comments on the previous version of their paper, and I do not have further objections against publication of this manuscript in PloS One.

Reviewer #2: The reason the authors consider the effect being cross modal is that the cue modality was found to affect target processing. In this case, what is cross modality should be "cue-induced target sensitivity" rather than the "distractor induced blindness". I still think that given the DIB phenomenon is centered on the effect of the distractor, the modality of interest should be between the distractor and the target, rather than the cue. This should, at the very least, be acknowledged as a limitation and discussed. The suboptimal operationalization in previous studies does not justify the continue of the conduct.

In the newly added contents, the authors mentioned several ERP components that only emerge in cross-modal DID experiments but not uni-modal experiments. One question is whether these effects are truly due to cross modality in general, or caused by the particular modality of a *visual* stimuli in addition to auditory materials in Kern and Niedeggen (2023). The present study served as a nice control condition to the previous study by reversing the modality of the cue and target, creating a fully crossed modality manipulation when viewed together with the previous DID study. However, given that the findings of the present research suggest an auditory cue does not alter the visual DIB effect, can we conclude that the previous finding of a visual cue altering the auditory DID is an effect unique to the visual modality of the cue, rather than a general cross-modality effect?

In addition, the cross-modal effect described by the authors--cue modality affects target identification--sounds like only a main effect, which is also likely well studied. Ideally, it would only make sense to vary cue validity in the current context if there has been initial evidence suggesting that cue validity modulates the effect of the distractors on target--an interaction between cue and distractor in DIB or DID, rather a simple main effect of the cue. I hope the authors make it clearer in the paper what kind of modulation effect the cue has.

My concern regarding the reliability given the small number of trials still holds. The authors mentioned that 55 trials were used per condition, and artifact rejection abandoned around 50% of the trials. (1) Please specify whether there were 55 trials before or after artifact rejection. If it is after rejection, only then less than 30 trials were available for analysis, which could be too few for reliable average waveforms. (2) It is unusual to see such a large proportion of trials abandoned in artifact rejections, making it questionable whether the remaining trials are truly representative of the overall performance. In other words, the effect may only occur for a selected subset of the trials after rejection. Please provide some clarification on why artifacts such as eye movements and head movements were not controlled during the experiment, but instead harshly rejected in later processing.

7. PLOS authors have the option to publish the peer review history of their article (what does this mean?). If published, this will include your full peer review and any attached files.

Reviewer #1: No

Reviewer #2: No

---

## [Author Response · Author response to Decision Letter 1]

1 Aug 2024

See "Response to Reviewers" document included in the upload.

---

## [Decision Letter · Decision Letter 2]

13 Aug 2024

Event-related potentials of stimuli inhibition and access in cross-modal distractor-induced blindness

PONE-D-24-10062R2

Dear Dr. Hanke,

We’re pleased to inform you that your manuscript has been judged scientifically suitable for publication and will be formally accepted for publication once it meets all outstanding technical requirements.

Kind regards,

Patrick Bruns

Academic Editor

PLOS ONE

Additional Editor Comments (optional):

Reviewers' comments:

Reviewer's Responses to Questions

**Comments to the Author**

1. If the authors have adequately addressed your comments raised in a previous round of review and you feel that this manuscript is now acceptable for publication, you may indicate that here to bypass the “Comments to the Author” section, enter your conflict of interest statement in the “Confidential to Editor” section, and submit your "Accept" recommendation.

Reviewer #2: All comments have been addressed

2. Is the manuscript technically sound, and do the data support the conclusions?

Reviewer #2: Yes

3. Has the statistical analysis been performed appropriately and rigorously? 

Reviewer #2: Yes

4. Have the authors made all data underlying the findings in their manuscript fully available?

Reviewer #2: Yes

5. Is the manuscript presented in an intelligible fashion and written in standard English?

Reviewer #2: Yes

6. Review Comments to the Author

Reviewer #2: The authors have now addressed both of my comments on the cross-modality nature of the DIB effect and the reliability of the data after harsh artifact rejection. Good to see the added discussion on the modality manipulation on the cue, and the explanation that attention had to split between two modalities helps illustrate the key rationale behind this manipulation. The additional ERP analysis based on more lenient artifact rejection also relieved my concern on the representativeness of the data to an extent. (The convention of artifact rejection is that no more than 25% of trials should be abandoned for each individual, to ensure that the trimmed data is representative of overall performance and to validate that participant was relatively engaged in the task.) Good to see that the original results could be replicated with the use of a larger subset of the EEG data.

7. PLOS authors have the option to publish the peer review history of their article (what does this mean?). If published, this will include your full peer review and any attached files.

Reviewer #2: No

---

## [Editor Report · Acceptance letter]

27 Aug 2024

PONE-D-24-10062R2 

PLOS ONE

Dear Dr. Hanke, 

I'm pleased to inform you that your manuscript has been deemed suitable for publication in PLOS ONE. Congratulations! Your manuscript is now being handed over to our production team.

Kind regards, 

on behalf of

Dr. Patrick Bruns 

Academic Editor

PLOS ONE